

# Global catalog of soil moisture droughts over the past four decades

Jan Řehoř[1,2], Rudolf Brázdil[1,2], Oldřich Rakovec[3,4], Martin Hanel[4], Milan Fischer[2,5], Rohini Kumar[3], Jan Balek[2,5], Markéta Poděbradská[2,5], Vojtěch Moravec[4,6], Luis Samaniego[3,7] and Miroslav Trnka[2,5]

[1]Institute of Geography, Masaryk University, Brno, 61137, Czech Republic
[2]Global Change Research Institute of the Czech Academy of Sciences, Brno, 60300, Czech Republic
[3]UFZ-Helmholtz Centre for Environmental Research, Leipzig, 04318, Germany
[4]Faculty of Environmental Sciences, Czech University of Life Sciences Prague, Praha, 16500, Czech Republic
[5]Department of Agrosystems and Bioclimatology, Mendel University in Brno, Brno, 63132, Czech Republic
[6]T. G. Masaryk Water Research Institute, Praha, 16000, Czech Republic
[7]University of Potsdam, Institute of Environmental Science and Geography, Am Neuen Palais 10, 14469 Potsdam, Germany

Correspondence to: Jan Řehoř (rehor.j@mail.muni.cz)

**Abstract.** At the global scale, droughts can be described by many variables, expressing their extent, duration, dynamics and severity. To identify common features in global land drought events (GLDEs) based on soil moisture, we present a robust method for their identification and classification (cataloging). Gridded estimates of root-zone soil moisture from the SoilClim model and the mesoscale Hydrologic Model (mHM) were calculated over global land from 1980–2022. Using the 10th percentile thresholds of soil moisture anomalies and OPTICS clustering of the gridded data in a 10-day interval, a total of 775 GLDEs from SoilClim and 630 GLDEs from mHM were identified. By utilizing four spatiotemporal and three motion-related characteristics for each GLDE, we established threshold percentiles based on their distributions. This information enabled us to categorize droughts into seven severity categories (ranging from extremely weak to extremely severe) and seven dynamic categories (ranging from extremely static to extremely dynamic). Our global-scale synthesis revealed the highest relative proportions of extremely severe and extremely dynamic GLDEs in the South American region, followed by North America, while the single longest and most extensive GLDEs occurred in Eurasia. The severity and dynamic categories overlapped substantially for extremely severe and extremely dynamic droughts but very little for less severe/dynamic categories, despite some very small droughts that have occasionally been very dynamic. The frequency of GLDEs has generally increased in recent decades across different drought categories but is statistically significant only in some cases. Overall, the cataloging of GLDEs presents a unique opportunity to analyze the evolving features of spatiotemporally connected drought events in recent decades and provides a basis for future investigations of the drivers and impacts of dynamically evolving drought events.

## 1 Introduction

Droughts represent an important natural hazard with significant consequences for several areas of human activities, such as agriculture, forestry, and water management. Recent warming may significantly increase the extent, duration and severity of drought events in commonly affected, dry areas of the world, as well as in parts of the world that were not often strongly



affected in the past (e.g., Trenberth et al., 2014; Naumann et al., 2018; IPCC, 2021). The frequent occurrence of severe droughts, particularly since the 2010s, in different parts of the world has greatly increased according to past studies that

analyzed such events (e.g., Shmakin et al., 2013; Van Dijk et al., 2013; Griffin and Anchukaitis, 2014; Erfanian et al., 2017; Ionita et al., 2017; Marengo et al., 2017; Spinoni et al., 2017, 2019; Deng et al., 2020; Chiang et al., 2021; Moravec et al., 2021; Rakovec et al., 2022; Liu et al. 2023; Arias et al., 2024; Garrido-Perez et al., 2024). In this context, the compound effects of droughts with heat waves have appeared to be a problem in recent years, as documented in many studies that analyzed their spatiotemporal variability at the global (e.g., Mukherjee et al., 2020; Mukherjee and Mishra, 2021; Afroz et al., 2023; Wang

et al., 2023) or continental (e.g., for Europe Bezak and Mikoš, 2020; Sutanto et al., 2020; Ionita et al., 2021) scale.

To understand the spatiotemporal variability and severity of droughts, meteorological drought indices have traditionally been used (e.g., Spinoni et al., 2014, 2015, 2019; Chiang et al., 2021; Fuentes et al., 2022; Vicente-Serrano et al., 2022), enabling longer time series to be analyzed due to their simplicity. Drought indices based on satellite observations provide a new opportunity to study both the climatic aspects of drought and the land surface response to drought in a spatially

explicit manner (e.g., Anderson et al., 2011; Khan and Gilani, 2021; Nikraftar et al., 2021). A relatively complex approach is soil moisture modeling through hydrological and land-surface models in drought studies. These models can either use outputs of general circulation models to predict future soil moisture changes (e.g., Berg and Sheffield, 2018; Trugman et al., 2018) or can be used in a retrospective mode, employing station-based data or reanalysis data as input (e.g., Kumar et al., 2013; Murray et al., 2023; Řehoř et al., 2024a). The main focus of these existing analyses is usually to assess long-term trends and locate

drying hotspots worldwide. However, their scope tends to be limited to basic statistical characteristics, as they either did not assess droughts as individual events at all or delimited and quantified drought events for predefined regions and did not study their actual spatiotemporal evolution as a tool for drought cataloging. Existing drought catalogs also do not use temporal resolutions finer than one month at the global scale; therefore, these catalogs have very limited potential in terms of understanding motion-related characteristics and studying the movement of individual drought events. Thus, there is great

potential for research that implements event-focused approaches employing high spatiotemporal resolution datasets to catalog droughts via the synthesis of various characteristics at the global scale, while few such catalogs exist at the regional scale, e.g., Moravec et al. (2019) or Camalleri et al. (2023). Such a catalog could also become a basis for subsequent studies investigating the drivers and impacts of dynamically evolving drought events.

The aim of this paper is to present a robust method for the detection and cataloging of droughts for global land by

combining different characteristics describing the occurrence, spatiotemporal evolution, intensity and dynamics of selected drought events. The methods of severity and dynamic classifications of global land drought events are applied to selected soil moisture variables calculated from the SoilClim model and the mesoscale Hydrologic Model for 1980–2022.



## 2 Data

### 2.1 SoilClim model

The SoilClim water balance model (Hlavinka et al., 2011; Trnka et al., 2020; Řehoř et al., 2021) is used to calculate the dynamics of plant-available soil moisture in four soil layers (0.0–0.1, 0.1–0.4, 0.4–1.0 and 1.0–2.0 m) by comparing the inflow and outflow water balance components. SoilClim accounts for soil water holding capacity and vegetation cover type, seasonal phenological development or leaf area index dynamics, and the model simulates root growth and snow cover (Trnka et al.,

2010; Řehoř et al., 2021). SoilClim was applied to each grid with a daily input of meteorological variables that consisted of precipitation, temperature at 2 m above the ground, dew-point temperature at 2 m, wind speed at 10 m, and incoming shortwave radiation, which originates from ERA5-Land (Muñoz-Sabater et al., 2021), as well as with the leaf area index (LAI), land use and terrain inputs. The plant-available water capacity of the soil was computed as the difference between the field capacity and wilting point based on the inputs from the SoilGrids database (Hengl et al., 2014, 2017). SoilClim reproduces changes in

long-term soil moisture dynamics in topsoil well (Trnka et al., 2015; Řehoř et al., 2024a). The calculated volumetric soil moisture was converted into relative available water (AWR), where 100% represents the full field capacity and 0% represents the wilting point (for more details, see Řehoř et al., 2024a). This study used data for the 2.0 m soil depth obtained by aggregating all four modelled layers. The prepared SoilClim dataset covers a global nonglaciated land with a 0.5° resolution, excluding latitudes above 72° N and all of Antarctica.

### 2.2 Mesoscale Hydrologic Model

The mesoscale Hydrologic Model (mHM; Samaniego et al., 2010; Kumar et al., 2013) simulates hydrological processes at the mesoscale. This model considers the complex interplay of land surface and subsurface properties through multiscale parameter regionalization using land cover, terrain, and soil characteristics, and it simulates major water fluxes, such as evaporation, infiltration, river runoff, or groundwater flow. This study considers soil moisture (SM) simulations averaged over the entire 2

m soil column depth (aggregating values over six soil layers) to quantify shallow water availability. The daily meteorological inputs (i.e., precipitation and air temperature) originate from the ERA5 reanalysis (Hersbach et al., 2020). The daily minimum, maximum, and mean temperatures are also used to obtain potential evapotranspiration estimates (Hargreaves and Samani, 1985). Our simulations are based on the existing model setup using the digital elevation model from the USGS (Danielson and Gesch, 2011), soil characteristics from SoilGrids (Hengl et al., 2017), land cover from the ESA (Arino et al., 2012) and LAI

climatology from NASA (Tucker et al., 2005). The mHM is one of the seven global hydrological models that the WMO has evaluated and used for the development of the annual State of Global Water Resources reports since 2011, focusing on > 500 major hydrological basins worldwide (WMO, 2022). The mHM dataset prepared for this study covers a global nonglaciated land with a 0.5° resolution, excluding latitudes above 72° N and all of Antarctica.





## 3 Methods

### 3.1 Preparation and clustering of the drought datasets

The daily AWR data obtained from the SoilClim model and the daily SM data from the mHM were aggregated into 10-day intervals. Then, 10th percentile thresholds for AWR and SM were calculated for each grid and each 10-day interval (by applying a 30-day window to smooth the annual variation) for the entire 1980–2022 period to represent drought conditions. Drought occurrence has been identified using the 10th-percentile drought, which is in line with using this threshold in US Drought Monitoring (Svoboda et al., 2002) since 1995 for the "D2" category definition and in the Czech Drought Monitor System (Trnka et al., 2020; Intersucho, 2024) since 2012 as the "S2" category. To assess the most severe drought, the 2nd-percentile drought (i.e., 50-year return period) was calculated using the same approach.

Both 10th-percentile drought datasets were further clustered by the "ordering points to identify the clustering structure" (OPTICS) method (Ankerst et al., 1999), which was applied to the whole three-dimensional dataset (spatiotemporal) covering 43 years. OPTICS is suitable for delimiting clusters of varying density and shape, without requiring the specification of the number of clusters beforehand. To eliminate regional cases with very small drought-affected areas, clusters that included fewer than 50 grids for one 10-day interval, fewer than 500 grids overall or that appeared in less than three 10-day intervals were excluded from both datasets, after which 775 clusters (further drought events) remained in the SoilClim dataset and 630 remained in the mHM dataset.

### 3.2 Severity classification of global land droughts

To describe the severity of the identified drought events, four spatiotemporal characteristics were calculated as follows:

(a) Maximum areal extent of a drought event during its duration.

(b) The total sum of the areal extent of drought event for all individual 10-day intervals during its duration.

(c) Duration of drought event at 10-day intervals.

(d) Drought intensity expressed as the total sum of areal extents of 2nd-percentile drought within the drought event during its duration.

Subsequently, for each of these characteristics and both datasets, the identified drought events (775 for SoilClim and 630 for mHM) were placed in order from the lowest to the highest values of the given characteristic, and orders of the values were used as scores. For example, a score of 1 was attributed to the event with the lowest value of the given characteristic, and a score of 775 or 630 was attributed to the event with the highest value of this characteristic for SoilClim and mHM, respectively. In cases where multiple events shared the same value for a given characteristic, their mean order was assigned to all of them. Then, for each drought event, sums of the scores of the four above characteristics were calculated and used for severity classification. These final scores were further divided based on percentile thresholds to classify individual events into the following seven severity categories (for each dataset) of global land droughts: 1s – extremely weak drought (<5th percentile), 2s – very weak drought (5th–20th percentile), 3s – weak drought (20th–35th percentile), 4s – average drought





(35th–65th percentile), 5s – severe drought (65th–80th percentile), 6s – very severe drought (80th–95th percentile), and 7s – extremely severe drought (>95th percentile).

## 3.3 Dynamic classification of global land droughts

For each identified drought event and each 10-day interval of its duration, centroids were calculated using the median of the

longitudes and latitudes for individual grids. Subsequently, three dynamic characteristics of these centroids for individual drought events were calculated as follows:

(a) The total sum of geographic distances between centroid positions for all individual 10-day intervals.

(b) The maximum distance between two centroid positions during the entire duration of a drought event.

(c) The mean geographic distance between centroid positions for all individual 10-day intervals.

Furthermore, analogous to the severity classification, for each of these characteristics and both datasets, the drought events were ordered from the lowest to the highest value of a given characteristic, and their orders were used as scores. Then, the sums of scores from all three characteristics of each identified drought event were divided by percentile thresholds to classify them according to SoilClim and mHM into the following seven dynamic categories: 1d – extremely static drought (<5th percentile), 2d – very static drought (5th–20th percentile), 3d – static drought (20th–35th percentile), 4d – drought of

average movement (35th–65th percentile), 5d – dynamic drought (65th–80th percentile), 6d – very dynamic drought (80th–95th percentile), and 7d – extremely dynamic drought (>95th percentile).

## 4 Results

### 4.1 Global land droughts according to the SoilClim model

### 4.1.1 Severity classification

By applying a clustering approach to gridded daily AWR data from SoilClim, a total of 775 global land drought events (GLDEs) from 1980–2022 were detected. For their severity classification, four severity characteristics and the methodology of their analysis, described in Sect. 3.2, were used. The complete list of all identified GLDEs is included in Table S1. Fig. 1 shows the interrelationships among the four basic characteristics used for severity classification. Although individual GLDEs exhibit significant spatiotemporal variability, there is a noticeable grouping pattern of GLDE categories in the scatterplots.

However, for less severe categories, the individual characteristics are variable, often compensating for each other; for extremely severe droughts (category 7s), all characteristics consistently display very high scores. Some plots of pairs from the four characteristics in Fig. 1 display the grouping of GLDEs into lines because there are many GLDEs with the same value (meaning the same score assigned to them) in the case of a single-digit number of 10-day intervals. The described features of the scatterplots in Fig. 1 are confirmed in the boxplots of these categories, as shown in Fig. 2, where extremely severe droughts

(7s) were characterized by the highest values of all box plot parameters (median, lower and upper quartile, maximum and





minimum), decreasing stepwise over the categories of very severe (6s) and severe droughts (5s) to average droughts (4s). The decreases in the four characteristics from average droughts over weak (3s) and very weak (2s) to extremely weak droughts (1s) were much smaller, as there were many more GLDEs with small spatiotemporal extents.



**Figure 1: Relationships between four basic severity characteristics of global land drought events (a – maximum extent, b – total extent, c – duration, d – intensity) according to SoilClim model divided into seven drought categories for the 1980–2022 period.**



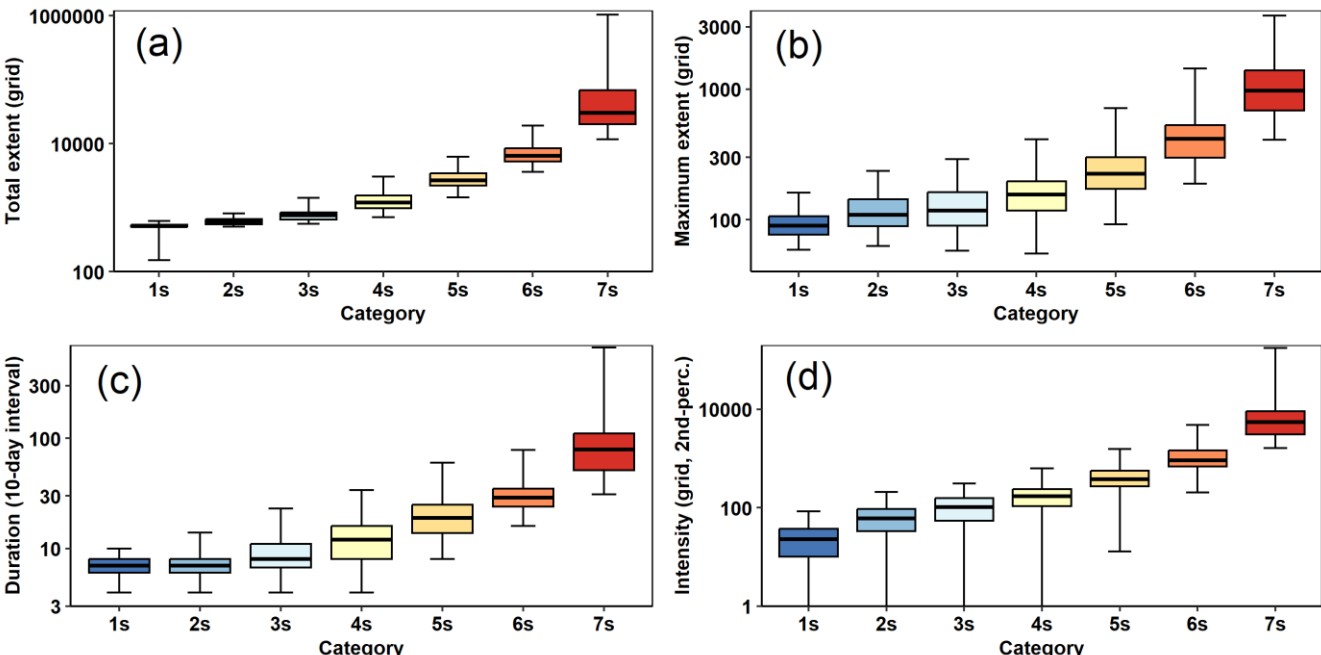

**Figure 2: Boxplots (median, upper and lower quartile, maximum and minimum) of four basic severity characteristics of global land drought events (a – maximum extent, b – total extent, c – duration, d – intensity) according to the SoilClim model divided into seven drought categories for the 1980–2022 period.**

Fig. 3 shows the distribution of GLDEs among individual continents, which corresponds well with the size of individual continents (Europe and Asia are considered together as Eurasia). Eurasia experienced the greatest number of GLDEs (337, 43.5% of all GLDEs), followed by North America (193/24.9%), Africa (115/14.8%), South America (82/10.6%) and Australia (48/6.2%) (Fig. 3a). However, concerning severity category 7s of the GLDEs, most occurred in North America (14), followed by Eurasia (12), while Eurasia was dominant in category 6s, with 57 detected GLDEs. In terms of the relative distribution across seven categories within a given continent (as shown in Fig. 3b), North America had the highest proportion of GLDEs in category 7s (7.3%), followed by South America and Africa (6.1%), while Australia had the highest proportion of GLDEs in category 6s (20.8%). Severe droughts (5s) had the highest proportion of 19.5% in South America. Extremely weak droughts (1s) accounted for 7.8% of the total drought in Africa, and very weak (2s) and weak (3s) droughts were the most frequently occurring in South America (19.5 and 17.1%, respectively).





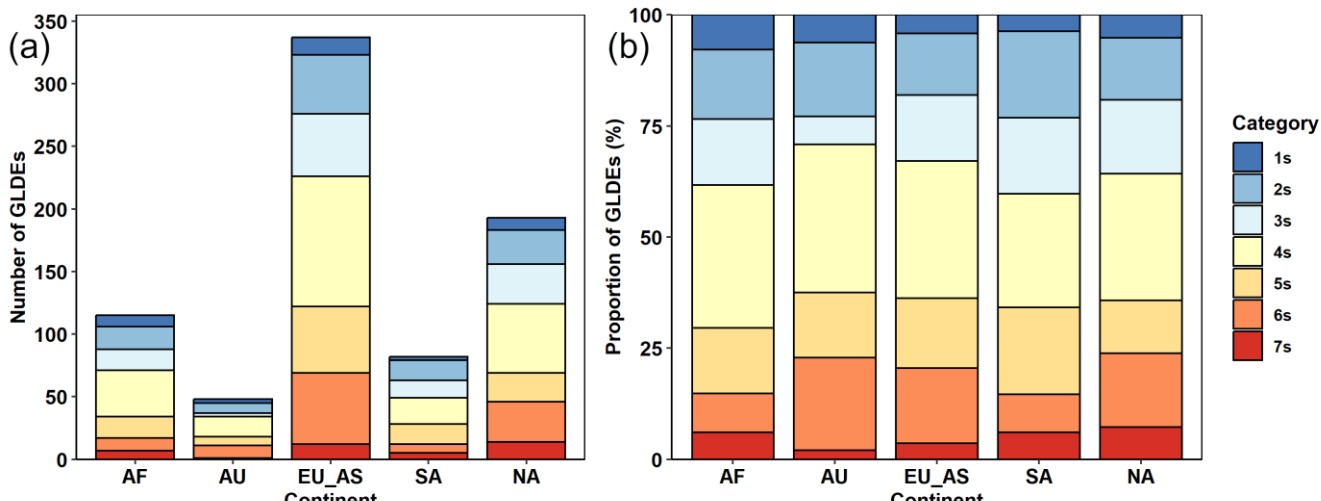

**Figure 3: Continental distribution of seven drought categories from the severity classification of global land drought events (GLDEs) according to the SoilClim model for 1980–2022: (a) total numbers of GLDEs; (b) relative proportions of seven categories of GLDEs for a given continent. Continents: AF – Africa, AU – Australia, EU_AS – Eurasia, SA – South America, NA – North America.**

Of the ten most extreme GLDEs selected according to drought severity classification, four occurred in Eurasia, three occurred in North America, two occurred in Africa and one occurred in South America (Table 1). Except for two GLDEs in Eurasia from July 1982–August 1988 and from June 1991–August 1994, the other eight appeared after 2000. For each of these four continents, one of these most extreme droughts was still ongoing in the last decade of the analyzed dataset, ending in 2022. The most extreme identified drought, which began in November 2004 in eastern Eurasia, was still ongoing until

December 2022 and achieved a maximum extent of 6.7 million km$^2$ in 2021. The second most extreme drought, which began in May 2019 and continued after 2022, occurred in North America and exceeded its maximum area of 8.5 million km$^2$ in 2020. According to the final scores, the third most common event was an event lasting from October 2014 to November 2019 in Africa; however, it had a maximum area of 4.8 million km$^2$ in 2019 behind the other drought in Africa starting in February 2020 and continuing after December 2022. The latter had a maximum extent of 5.9 million km$^2$ in 2022.


**Table 1: Ten most extreme global land drought events (based on severity scores) from the SoilClim model according to the severity classification. The severity characteristics are specified according to points a–d in Sect. 3.2 (* indicates ongoing droughts).**

| Max. area (km$^2$) | Duration | No. of 10-day intervals | Continent | Severity scores | | | | |
|---|---|---|---|---|---|---|---|---|
| | | | | a | b | c | d | Total |
| 6 701 638 | Nov/04–Dec/22* | 662 | Eurasia | 775 | 775 | 775 | 775 | 3100 |
| 8 522 546 | May/19–Dec/22* | 132 | S. America | 774 | 774 | 769 | 774 | 3091 |
| 4 781 107 | Oct/14–Nov/19 | 188 | Africa | 768 | 773 | 773 | 768 | 3082 |
| 5 923 178 | Feb/20–Dec/22* | 107 | Africa | 772 | 771 | 764 | 773 | 3080 |
| 2 532 446 | Jul/99–Jun/04 | 180 | N. America | 759 | 772 | 771 | 771 | 3073 |





| 3 492 783 | Jun/02–Nov/04 | 88 | Eurasia | 773 | 767 | 759 | 767 | 3066 |
|---|---|---|---|---|---|---|---|---|
| 3 579 831 | Aug/10–Jun/13 | 104 | N. America | 766 | 766 | 763 | 765 | 3060 |
| 1 820 395 | Jun/91–Aug/94 | 116 | Eurasia | 761 | 765 | 767 | 766 | 3059 |
| 2 722 325 | Jul/82–Aug/88 | 223 | Eurasia | 755 | 769 | 774 | 760 | 3058 |
| 2 850 866 | Aug/20–Dec/22* | 88 | N. America | 764 | 763 | 759 | 772 | 3058 |

### 4.1.2 Dynamic classification

Fig. 4 shows the interrelationships among the three characteristics of the dynamic drought classification of 775 GLDEs calculated from the SoilClim model for 1980–2022, as described in Sect. 3.3. Compared to the four characteristics of severity classification (Fig. 1), they show more consistent patterns with more concentrated fields of related points, particularly for categories that include GLDEs with average movements (category 4d) to extremely static droughts (1d). The box plots of these categories are shown in Fig. 5, which reveals that the employed characteristics decrease in a stepwise manner from the category

of extremely dynamic droughts (7d) to that of extremely static droughts (1d), with nearly no overlap in values among the interquartile ranges of the seven individual categories.



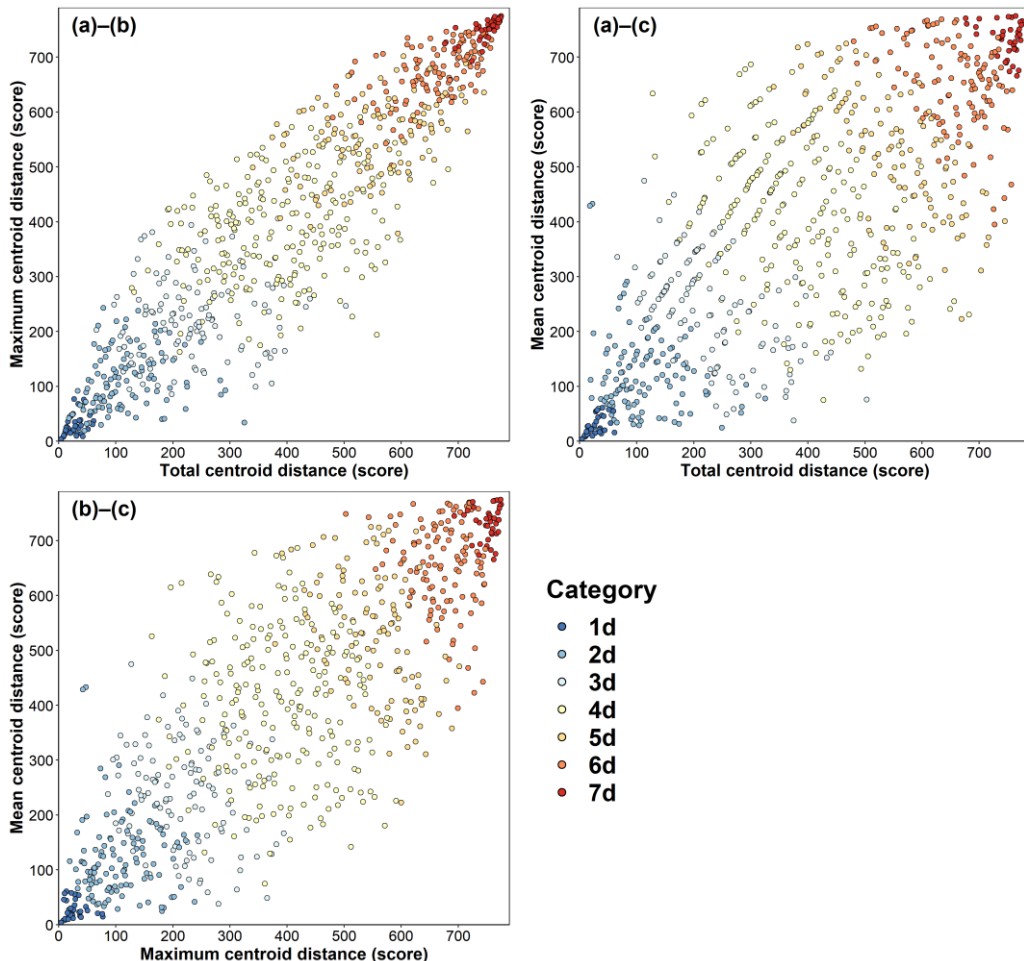

**Figure 4: Relationships between three basic dynamic characteristics of global land drought events (a – total centroid distance, b – maximum centroid distance, c – mean centroid distance) according to the SoilClim model divided into seven drought categories for 1980–2022.**



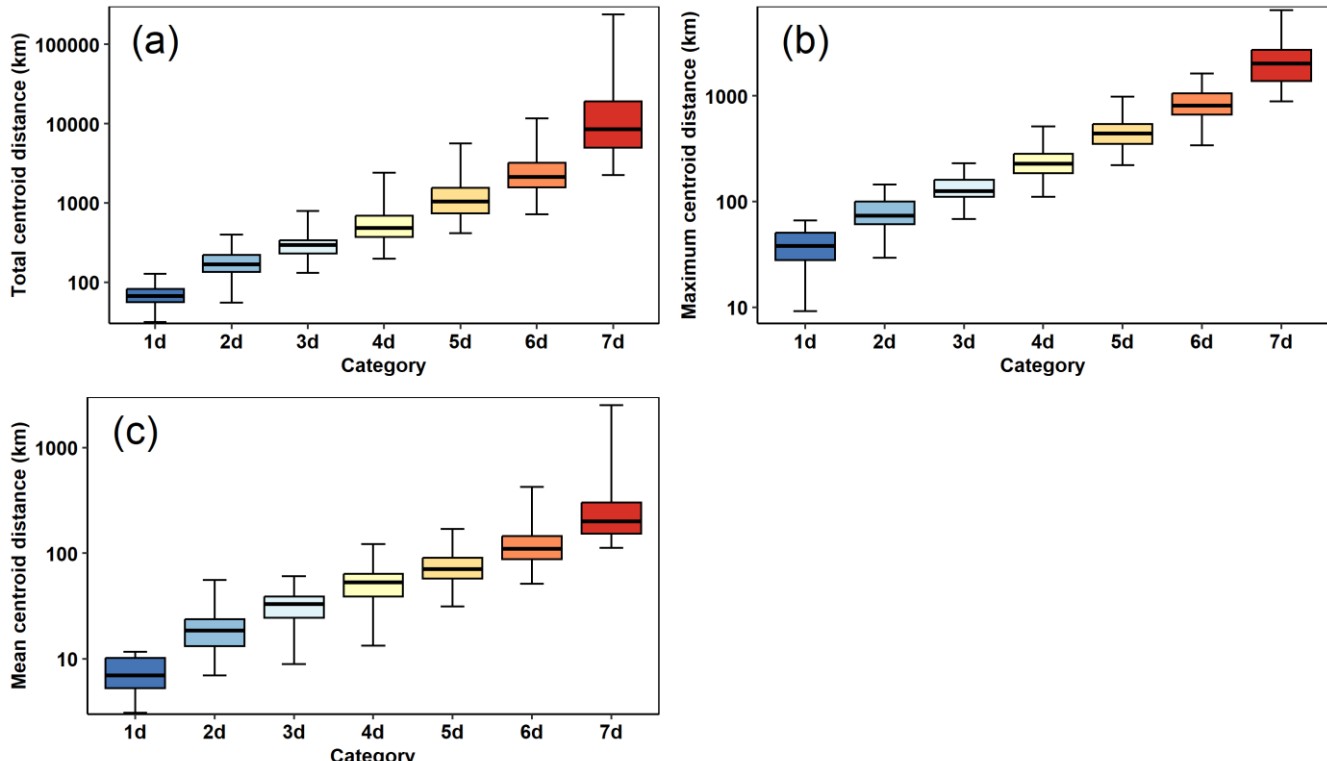

**Figure 5: Box plots of the three basic dynamic characteristics of global land drought events (a – total centroid distance, b – maximum centroid distance, c – mean centroid distance) according to the SoilClim model divided into seven drought categories for 1980–2022.**

For the overall distribution of GLDEs among individual continents (Fig. 6a), the total numbers correspond to those shown in

Fig. 3a but with different numbers of categories according to the dynamic classification. Extremely dynamic GLDEs (category 7d) occurred on all continents, with a maximum of 11 events occurring in North America, followed by 9 events occurring in Africa. Very dynamic GLDEs (category 6d) were the most frequently occurring in Eurasia, with 47 events. Concerning the relative proportions of seven dynamic categories on a given continent (Fig. 6b), extremely dynamic droughts (7d) had the highest proportions in South America (8.6 %), very dynamic droughts (6d) had the highest proportion in North America

(15.8 %), while dynamic droughts (5d) had the highest relative proportions in Africa (16.7 %). Concerning static droughts, 7.7 % were extremely static (1d) in Eurasia, 18.7 % were very static (2d) in North America and 16.6 % were static (3d) in Eurasia and North America.



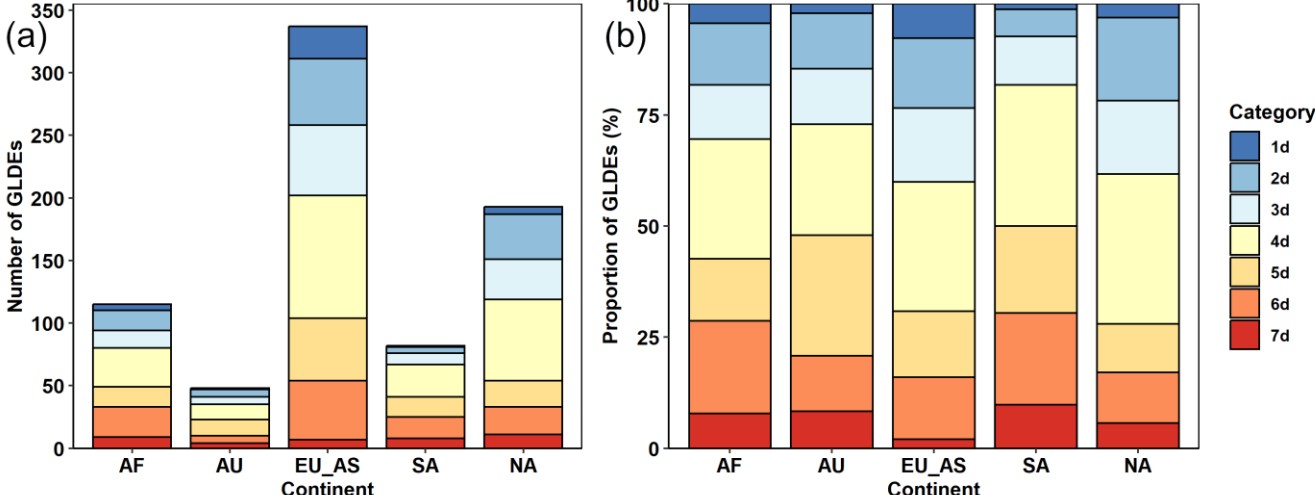

**Figure 6: Continental distribution of seven drought categories from the dynamic classification of global land drought events (GLDEs) according to the SoilClim model for 1980–2022: (a) total numbers of GLDEs; (b) relative proportions of seven categories of GLDEs for a given continent. Continents: AF – Africa, AU – Australia, EU_AS – Eurasia, SA – South America, NA – North America.**

Although all ten selected extreme GLDEs according to severity classification (Table 1) belonged concurrently with the category of extremely dynamic drought (7d), only six events appeared among the ten most dynamic droughts (Table 2). The long-term drought event in Eurasia that spanned from November 2004 to December 2022 exhibited a mean centroid movement of 360 km, followed by 347 km for a drought event from December 2013 to April 2016 in South America and 309 km for drought from October 2014 to November 2019 in Africa. The three most dynamic droughts occurred in Africa and South America, and two occurred in Eurasia and North America.

**Table 2: Ten most extreme global land drought events (based on the dynamic scores) from the SoilClim model according to dynamic classification. Dynamic characteristics are specified according to points a–c in Sect. 3.3 (* indicates ongoing droughts).**

| Max. area (km²) | Duration | No. of 10-day intervals | Continent | Dynamic scores | | | |
|---|---|---|---|---|---|---|---|
| | | | | a | b | c | Total |
| 6 701 638 | Nov/04–Dec/22* | 662 | Eurasia | 775 | 775 | 766 | 2316 |
| 5 867 390 | Dec/13–Apr/16 | 86 | S. America | 772 | 773 | 764 | 2309 |
| 4 781 107 | Oct/14–Nov/19 | 188 | Africa | 774 | 772 | 759 | 2305 |
| 197 279 | Jul/02–Oct/02 | 11 | Eurasia | 757 | 764 | 773 | 2294 |
| 5 923 178 | Feb/20–Dec/22* | 107 | Africa | 770 | 769 | 751 | 2290 |
| 4 320 892 | Apr/08–Apr/11 | 111 | S. America | 767 | 771 | 738 | 2276 |
| 8 522 546 | May/19–Dec/22* | 132 | S. America | 771 | 763 | 741 | 2275 |
| 2 532 446 | Jul/99–Jun/04 | 180 | N. America | 773 | 766 | 731 | 2270 |
| 3 579 831 | Aug/10–Jun/13 | 104 | N. America | 768 | 755 | 746 | 2269 |
| 2 635 026 | Feb/09–Mar/10 | 40 | Africa | 755 | 753 | 750 | 2258 |



### 4.1.3 Comparison of severity and dynamic classifications

To compare the distribution of GLDEs according to seven severity and dynamic classification categories, Fig. 7a shows the number of dynamic droughts that were attributed to the severity classification categories. The strongest relationship existed in category 7s, because 56.4% of events from this severity category belonged to dynamic category 7d and 35.9% were in category 6d (Fig. 7b), while no dynamic category lower than 5d appeared among 7s GLDEs. Category 6s coincided best with category 6d in 45.7% of events and with category 5d in 29.3%. High agreement between the categories was also found for 4s and 4d,

with 44.6%, which was much lower for 2s with 2d (25.9%). In the case of other drought severity categories, the agreement was the highest with any neighboring category (1s with 2d at 33.3%, and 1s with 3d at 30.8%; 3s with 4d at 29.3%, and 3s with 2d at 26.7%; 5s with 4d at 35.3%, and 5s with 5d at 28.4%). Categories 7d and 6d of GLDEs rarely appear within the lower severity categories; however, their presence indicates an asymmetric relationship, where most GLDEs are never totally static, while those with small extents are rarely extremely dynamic.


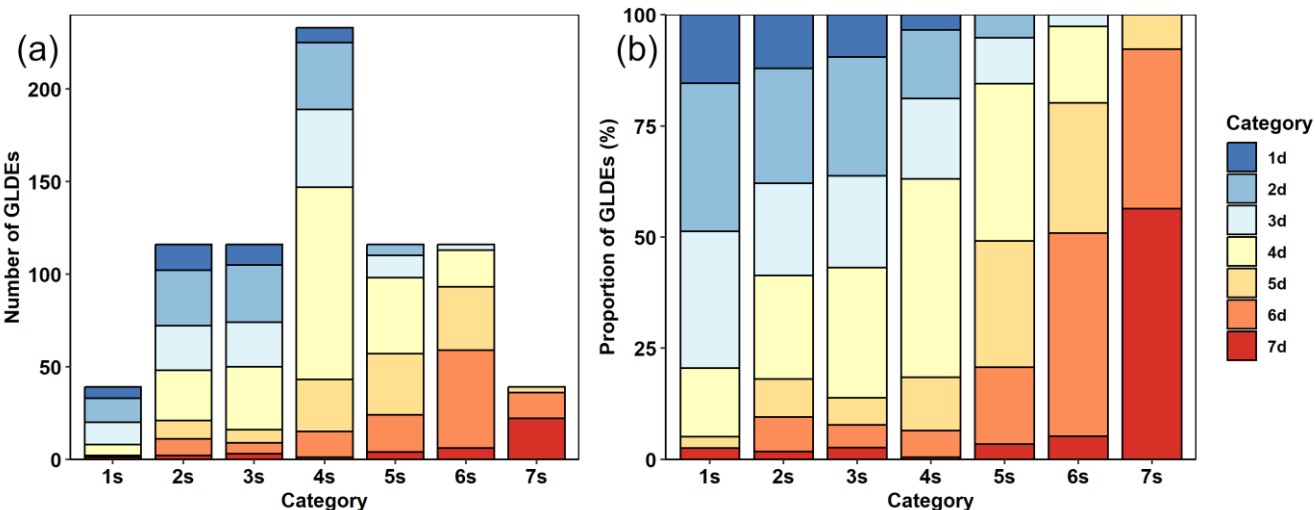

**Figure 7. Comparison of global land drought events (GLDEs) from severity and dynamic classifications from 1980–2022 according to the SoilClim model: (a) total numbers of dynamic categories in severity categories; (b) relative proportions of dynamic categories in severity categories.**

## 4.2 Global land droughts according to the mHM

### 4.2.1 Severity classification

Based on the four severity characteristics and the methodology of their analysis described in Sect. 3.2, a total of 630 GLDEs according to gridded daily SM data from the mHM were detected from 1980–2022 (Table S2). The relationships among the four basic characteristics shown in Fig. 8 are very similar to those shown in Fig. 1 based on the SoilClim model, as they are





described in Sect. 4.1.1. Box plots of these characteristics for seven drought categories (Fig. 9) show a clear stepwise decline
      from extremely severe drought (7s) to the category of average drought (4s), while three categories of weak drought (from 3s
      to 1s) show generally smaller differences and variability, except for drought intensity.



**Figure 8: Relationships between the four basic severity characteristics of global land drought events (a – maximum extent, b – total
      extent, c – duration, d – intensity) according to the mHM divided into seven drought categories for 1980–2022.**





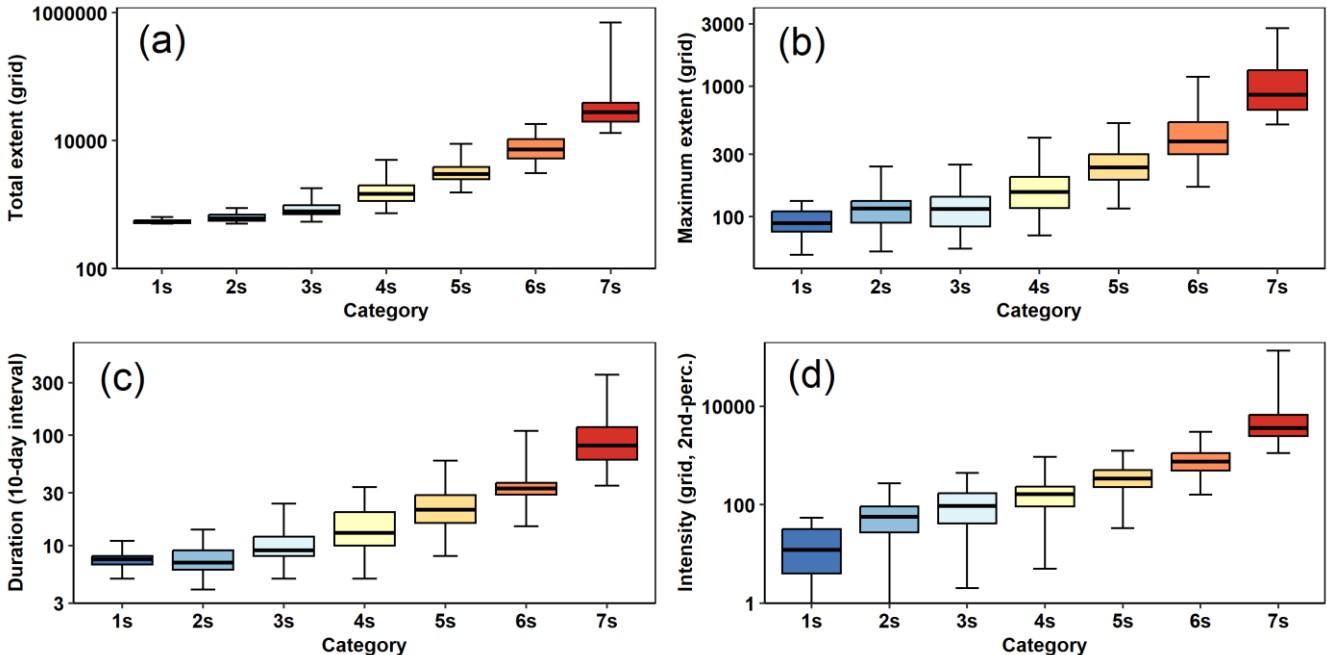

**Figure 9: Box plots of the four basic severity characteristics of global land drought events (a – maximum extent, b – total extent, c –**
**duration, d – intensity) according to the mHM divided into seven drought categories for 1980–2022.**

Considering the continental distribution (Fig. 10a), the predominance of Eurasia with 298 GLDEs, which accounts
for 47.3% of all events, appears to be slightly more pronounced than in the SoilClim data (cf. Fig. 3a), followed by North
America with 152 events (24.1%). Only 50 GLDEs (7.9%) were detected in Australia, which is less than in South America
(58 events and 9.2%) and Africa (72 events and 11.4%). Concerning the relative proportions of seven categories over each
continent, the maximum proportions are as follows (Fig. 10b): 1s, 3s and 5s – Africa (6.9, 20.8 and 16.7% respectively), 2s –
22.0% in Australia, 4s and 6s – North America (35.5 and 15.8%), and 7s – 8.6% in South America.



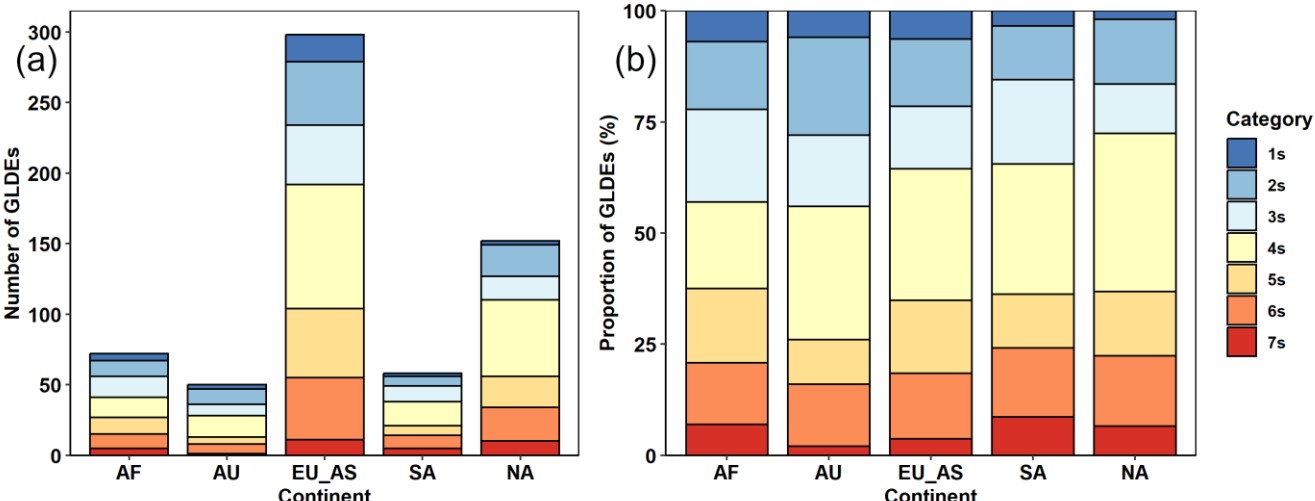

**Figure 10: Continental distribution of seven drought categories from the severity classification of global land drought events (GLDEs) according to the mHM for 1980–2022: (a) total numbers of GLDEs; (b) relative proportions of seven categories of GLDEs for a given continent. Continents: AF – Africa, AU – Australia, EU_AS – Eurasia, SA – South America, NA – North America.**

Among the ten most extreme GLDEs based on the severity drought classification from the mHM-based analysis (Table 3), three occurred in Eurasia and North America, two occurred in South America, one occurred in Africa and one occurred in Australia. Except for two GLDEs in North America from January 1980–May 1982 and October 1987–July 1991, the other eight events appeared after 2000. For each continent except for Australia, one of these droughts was still ongoing in the last decade of the analyzed dataset, ending in 2022. The most extensive and still ongoing drought, which began in June 2013 in eastern Eurasia, reached a maximum area of 9.5 million km$^2$ in 2021. The second most extreme GLDEs in Africa, which began in April 2013 and continue after December 2022, with a maximum area of 7.3 million km$^2$ in 2022, were exceeded by 7.6 million km$^2$ in 2020 by ongoing GLDEs in South America that began in December 2018. According to the scores, the third most extreme event occurred between April 2007 and July 2014 in Eurasia and reached a maximum extent of 4.8 million km$^2$ in 2011.

**Table 3: The ten most extreme global land drought events (based on severity scores) from the mHM according to severity classification. The severity characteristics are specified according to points a–d in Sect. 3.2 (* indicates ongoing droughts).**

| Max. area (km$^2$) | Duration | No. of 10-day intervals | Continent | Severity scores | | | | |
|---|---|---|---|---|---|---|---|---|
| | | | | a | b | c | d | Total |
| 9 539 417 | Jun/13–Dec/22* | 347 | Eurasia | 630 | 630 | 629 | 630 | 2519 |
| 7 292 765 | Apr/13–Dec/22* | 355 | Africa | 627 | 629 | 630 | 629 | 2515 |
| 4 761 254 | Apr/07–Jul/14 | 264 | Eurasia | 629 | 628 | 628 | 627 | 2512 |
| 7 629 586 | Dec/18–Dec/22* | 147 | S. America | 628 | 627 | 626 | 628 | 2509 |
| 3 013 866 | Oct/19–Dec/22* | 118 | N. America | 622 | 626 | 622 | 626 | 2496 |



| 3 307 609 | Oct/87–Jun/91 | 135 | N. America | 623 | 624 | 625 | 622 | 2494 |
| 2 294 400 | Dec/17–Jan/22 | 150 | Australia | 613 | 625 | 627 | 625 | 2490 |
| 2 087 318 | Jan/80–May/82 | 87 | N. America | 619 | 622 | 615 | 624 | 2480 |
| 5 855 938 | Jan/14–Apr/16 | 85 | S. America | 626 | 619 | 614 | 621 | 2480 |
| 3 297 702 | Jul/14–Feb/16 | 59 | Eurasia | 624 | 623 | 599 | 617 | 2463 |

### 4.2.2 Dynamic classification

The relationships among the three dynamic classification characteristics of the 630 GLDEs calculated from the mHM for
1980–2022 (Fig. 11) reveal relatively consistent patterns. There are well-distinguished fields of points that belong to extremely
dynamic (7d) and very dynamic (6d) GLDEs, and on the other hand, extremely static (1d) and very static (2d) events are
distinguished from the remaining categories. All seven categories of dynamic drought were well distinguished, as shown in
the box plots created for the three basic characteristics (Fig. 12).



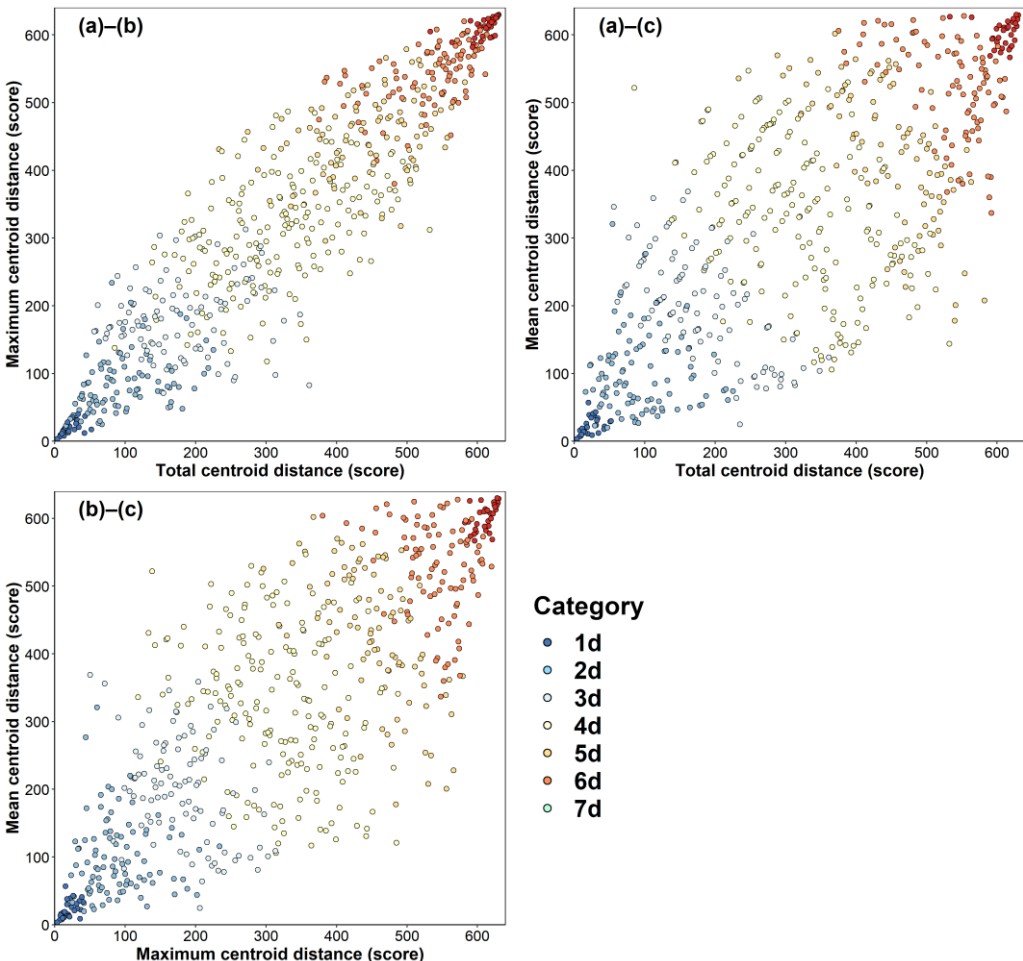


**Figure 11: Relationships between the three basic dynamic characteristics of global land drought events (a – total centroid distance, b – maximum centroid distance, c – mean centroid distance) according to the mHM divided into seven drought categories for 1980–2022.**





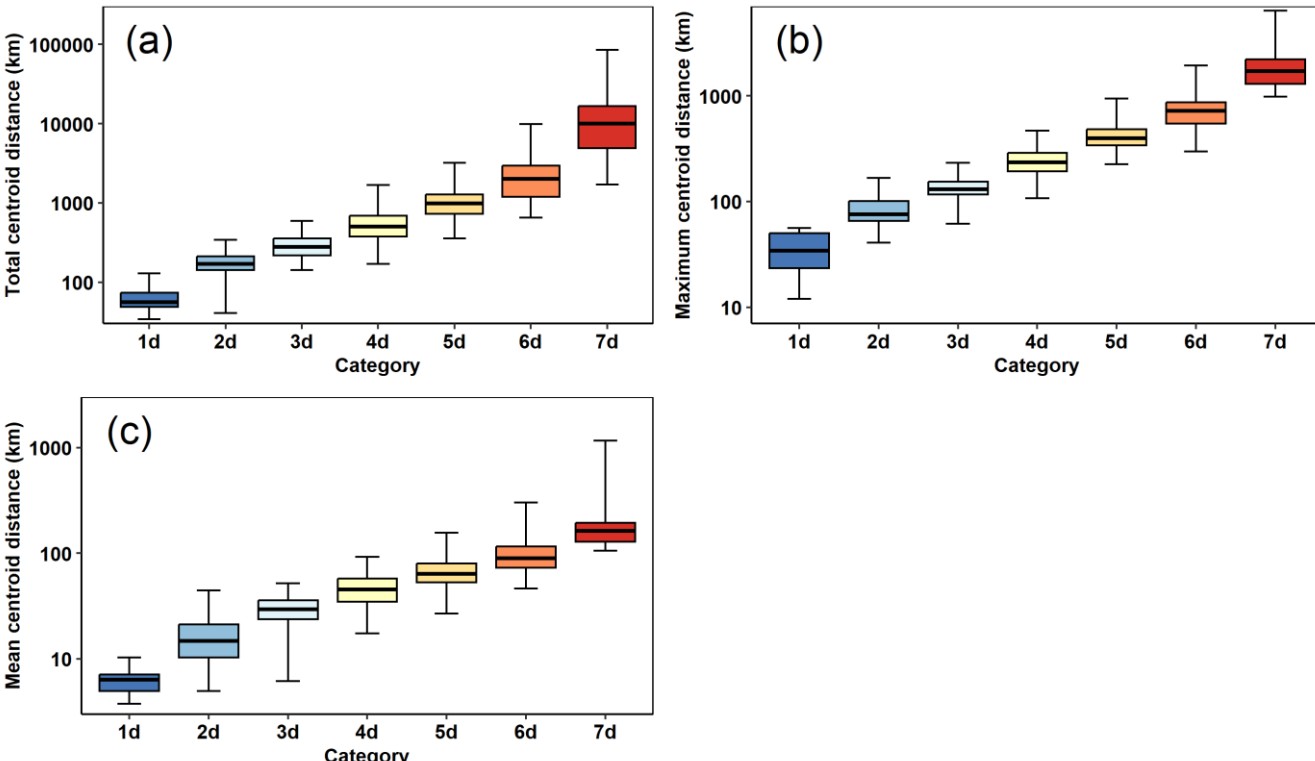


**Figure 12: Box plots of the three basic dynamic characteristics of global land drought events (a – total centroid distance, b – maximum centroid distance, c – mean centroid distance) according to the mHM divided into seven drought categories for 1980–2022.**

Concerning the total distribution of GLDEs among individual continents (Fig. 13a), their total numbers correspond to those in Fig. 10a but with different numbers of categories according to the dynamic classification. Extremely dynamic GLDEs (category 7d) occurred on all continents, with a maximum of 8 events occurring in South America and Eurasia. Very dynamic GLDEs (category 6d) were the most frequently occurring in Eurasia, with 36 events. Concerning the relative proportions of seven dynamic categories on a given continent (Fig. 6b), extremely dynamic droughts (7d) and very dynamic

droughts (6d) had the highest relative proportions in South America (13.8% and 27.6%, respectively), while dynamic droughts (5d) had the highest proportions in Australia (20.0%). Static droughts were most common in North America (7.9% for category 1d and 18.4% for category 2d) and South America (22.4% for category 3d).



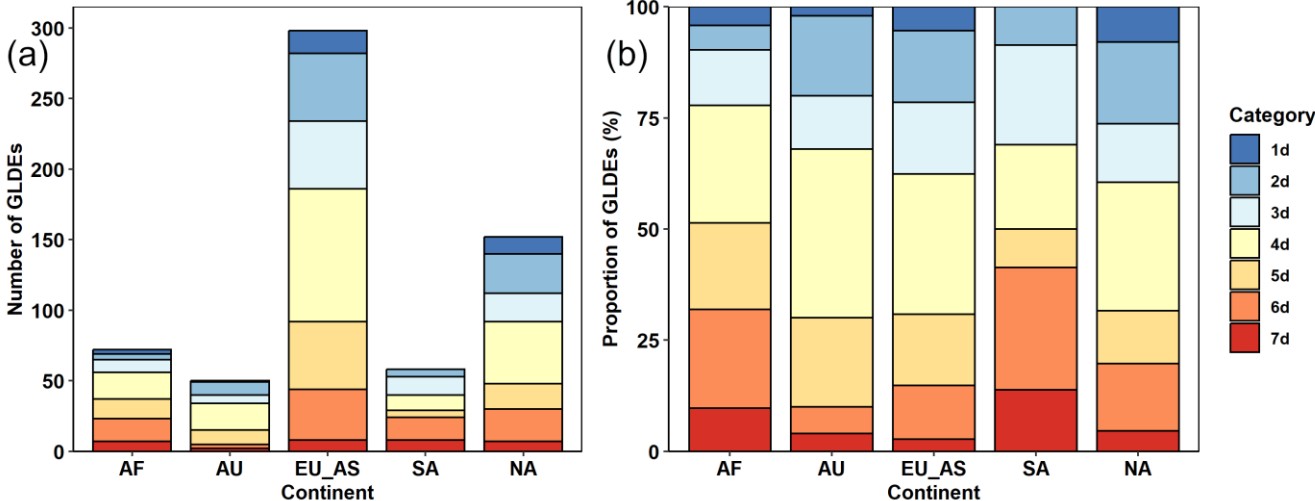

**Figure 13: Continental distribution of the seven drought categories from the dynamic classification of global land drought events (GLDEs) according to the mHM for 1980–2022: (a) total numbers of GLDEs; (b) relative proportions of the seven categories of GLDEs for a given continent. Continents: AF – Africa, AU – Australia, EU_AS – Eurasia, SA – South America, NA – North America.**

Although all ten of the most extreme GLDEs from the severity classification shown in Table 3 belonged concurrently to extremely dynamic drought, only six of these events appeared among the ten most dynamic droughts shown in Table 4. Out of these GLDEs, five cases were in Eurasia, three in South America and only one belonged to North America and Africa. The drought event in Eurasia from April 2007 to July 2014, with a mean centroid movement of 326 km, was classified as being the most dynamic. The second most dynamic event was classified between June 2014 and March 2015 in North America, and the third from June 2013 to December 2022 in Eurasia.

**Table 4: The ten most extreme global land drought events (based on the dynamic scores) from the mHM according to the dynamic classification. Dynamic characteristics are specified according to points a–c in Sect. 3.3 (* indicates ongoing droughts).**

| Max. area (km²) | Duration | No. of 10-day intervals | Continent | Dynamic scores | | | |
|---|---|---|---|---|---|---|---|
| | | | | a | b | c | Total |
| 4 761 254 | Apr/07–Jul/14 | 264 | Eurasia | 630 | 630 | 629 | 1889 |
| 334 297 | Jun/14–Mar/15 | 27 | N. America | 627 | 628 | 630 | 1885 |
| 9 539 417 | Jun/13–Dec/22* | 347 | Eurasia | 629 | 629 | 623 | 1881 |
| 2 715 481 | Mar/16–Feb/18 | 72 | S. America | 623 | 623 | 624 | 1870 |
| 7 629 586 | Dec/18–Dec/22* | 147 | S. America | 626 | 625 | 617 | 1868 |
| 5 855 938 | Jan/14–Apr/16 | 85 | S. America | 622 | 627 | 619 | 1868 |
| 7 292 765 | Apr/13–Dec/22* | 355 | Africa | 628 | 626 | 613 | 1867 |
| 3 300 973 | May/18–Aug/19 | 47 | Eurasia | 615 | 624 | 620 | 1859 |
| 2 306 091 | Mar/75–Jul/88 | 122 | Eurasia | 624 | 618 | 608 | 1850 |
| 3 297 702 | Jul/14–Feb/16 | 59 | Eurasia | 616 | 614 | 612 | 1842 |





### 4.2.3 Comparison of severity and dynamic classifications

A comparison of the GLDEs calculated from the mHM and divided according to seven severity and dynamic classification categories, as shown in Fig. 14, yields similar results to those obtained with the SoilClim model, as shown in Fig. 7. The highest agreement among the corresponding categories appears for extremely severe droughts 7s (56.3% with 7d and 40.6% with 6d) and very severe droughts 6s (40.4% with 6d and 25.5% with 5d) (Fig. 14b). Except the categories of average droughts 4s and 4d with a 35.1% agreement, the other severity drought categories showed better coincidence with neighboring categories: 1s with 2d at 25.0%, 2s with 4d at 30.2%, 3s with 4d at 34.4%, and 5s with 4d at 34.7%. The figure shows that categories from 1s to 5s overlap with droughts of average movement (4d). Category 1s overlapped significantly with the very dynamic GLDEs of category 6d (15.6%).

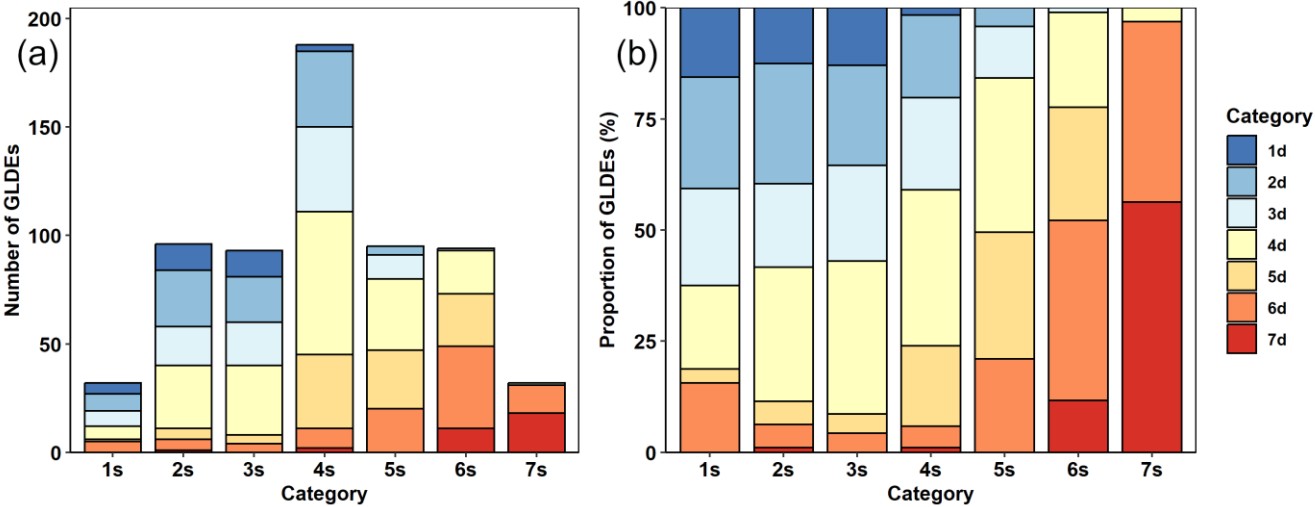

**Figure 14: Comparison of global land drought events (GLDEs) from severity and dynamic classifications from 1980–2022 according to the mHM: (a) total numbers of dynamic categories in severity categories; (b) relative proportions of dynamic categories in severity categories.**

### 4.3 Comparison of SoilClim and mHM droughts

With respect to the different numbers of detected GLDEs from SoilClim and mHM, the comparison relies on the relative proportions of seven severity (1s–7s) and dynamic (1d–7d) categories across individual continents (Fig. 15a,b). Additionally, the comparison extends to the proportions of dynamic categories 1d–7d in severity categories 1s–7s (Fig. 15c). In some cases, differences in relative proportions are very small; in others, they appear to be larger visually. For this reason, we used the two-



proportion Z test (Sprinthall, 2011) to test their statistical significance. Notably, statistically significant differences were observed only for category 4s in Africa and 1d in North America (both $p < 0.10$) and for proportions of dynamic droughts in severity categories for categories 1s and 6d ($p < 0.05$) and 4s and 4d ($p < 0.10$). These results indicate that outputs from different models generally agree regarding the continental distribution of delimited categories and connections between both classifications within a certain variability range partially connected to smaller sample sizes of GLDEs within individual

continents/categories.



**Figure 15:** Comparison of the relative proportions of global land drought events (GLDEs) based on SoilClim (SC) and mHM: (a) continental proportions of the seven drought categories 1s–7s according to severity classification; (b) continental proportions of seven drought categories 1d–7d according to dynamic classification; and (c) proportions of dynamic categories 1d–7d in severity categories 1s–7s. Continents: AF – Africa, AU – Australia, EU_AS – Eurasia, SA – South America, NA – North America.





## 5 Discussion

### 5.1 Drought classification and uncertainties in the results

The primary sources of uncertainties within the drought classifications presented here originate from the AWR/SM datasets, which are connected to either the models' algorithms or the input data. Because both models used meteorological inputs from reanalyses (ERA5-Land and ERA5), inherent biases must be considered, particularly in terms of their precipitation and temperature data. As Cucchi et al. (2020) discovered, ERA5 (including the ERA5-Land dataset) generally indicates an underestimation of temperature and an overestimation of precipitation at high elevations above sea level. This underestimation
could impact the delimitation of drought occurrence in these regions. Nevertheless, we argue that when using percentile transformation, drought characteristics are more robust against systematic biases in the absolute values of variables, such as temperature and precipitation, and the methodology enables us to capture the relative severity of drought characteristics.

Despite a possible lack of precision within the input data, some uncertainties are also intrinsic to the employed models. In the SoilClim model, there are uncertainties related to the schematization of individual processes, soil profile vertical discretization,
and other scale-related assumptions (e.g., each grid cell is represented by dominant landcover). These simplifications are inherent in any large-scale modeling scheme, and the uncertainty they create must be acknowledged. However, as we employed the AWR data in the form of percentile-based soil moisture anomalies, we were able to eliminate the effect of uncertainties that affect the long-term mean AWR within individual grids.

Furthermore, in both models, uncertainty originates from the underlying data, such as soil properties and land
cover/type characteristics, which must be approximated due to limited observational data. We also acknowledge that the fixed/prescribed model functions representing hydrological processes might not be equally plausible across different parts of the world.

Some uncertainties are associated with the clustering technique used, i.e., density-based clustering OPTICS. As described by Ankerst et al. (1999), objects are connected to form clusters if there is a sufficient number of other objects nearby
within a defined multidimensional space (in our case, three-dimensional). However, specific settings of the algorithm parameters must be partially derived empirically to fit the characteristics of a given dataset (particularly concerning its density) and prevent either the connection of all objects (in our case, grids in individual 10-day intervals) into one large cluster, including the whole dataset, or the failure of the algorithm to create clusters at all if the clustering parameters are too strict. There is no objective method for defining "perfect" parameters; hence, clustering uncertainty is inherent and affects the length
of existence of individual clusters in our dataset. A good example of this is the massive drought that occurred in Eurasia, classified as an extremely long GLDE from 2004 to 2022 (and still ongoing) from SoilClim, whereas using mHM data, it was separated into two GLDEs that were disconnected in 2013/2014 (cf. Tables 1 and 3). Potential alternatives for drought clustering have been proposed for example by Andreadis and Lettenmaier (2006), Vidal et al. (2010) or Samaniego et al. (2013).





Moreover, in cases of quickly varying densities of drought-affected grids and rapid spatial changes, some smaller, volatile clusters occasionally formed. These clusters may show questionably large values of its centroid's movement, which may be viewed as an artifact of the employed methods. Such an example could be the drought event from June 2014–March 2015 in North America (Table 4), which appeared around the center of the continent between other, more consolidated clusters. A few problematic cases such as this represent some uncertainty in the classification integrity. However, the selection of

multiple characteristics for both severity and dynamic classifications can partially mitigate the impact of such anomalous GLDEs. Another issue might be the arbitrary selection of percentile thresholds that were used to delimit seven categories in both classifications, with the particular aim to divide the extremes into smaller categories. However, slight changes in the thresholds would not alter the overall picture presented in this paper because the transition between categories is gradual, as shown in Figs. 1, 4, 8 and 11.

**5.2 A broader context for the classification results of cataloging droughts**

Numerous studies have explored different means of classifying global land droughts, employing a range of data, parameters, and methodologies. These studies typically present their findings through related global-scale maps. For example, Carrão et al. (2016) mapped global drought risk based on drought hazard, exposure, and vulnerability during 2000–2014. Spinoni et al. (2019) used the SPI and SPEI (at a scale ranging from 3 to 72 months) from 1951–2016 for 23 macroregions of the world to

obtain c. 4800 (SPEI-3) and 4500 (SPI-3) drought events. These events were divided into moderate, severe and exceptional events based on the drought severity, intensity, area, top event, peak intensity and area scores. He et al. (2020) developed a drought and flood catalog (GDFC) that spanned from 1950 to 2016. To define drought, they used both the SPI and soil moisture percentiles based on the VIC land surface model output. In this dataset, they employed simple clustering that connected neighboring grids to define individual drought episodes. They subsequently studied the relationship between drought area and

the intensity and return periods of severe droughts. Monjo et al. (2020) climatologically classified the duration of drought worldwide based on the dry–wet spell n-index derived from the global gridded daily Multi-Source Weighted-Ensemble Precipitation (MSWEP) dataset for 1979–2016. By comparing the different relationships between the occurrence of dry and wet spells, they identified seven types and presented their worldwide distributions. Fuentes et al. (2022) mapped spatiotemporal drought propagation through different subsystems at the global scale over recent decades using different standardized drought

indices. Drought propagation was established as the lag in the peak correlations between drought time series in different subsystems (for drought propagation, see, e.g., Li et al., 2023 for the Yellow River basin).

The classification approach for GLDEs presented in our study (Sects. 3.2 and 3.3) is not focused only on mapping any static drought state at the global scale, as in some of the above-reported studies, but rather, we present new views on the spatiotemporal variability and dynamics of drought events, combining several characteristics that describe their extent,

duration, severity and propagation. This research suggests a complex approach for monitoring droughts as individual dynamic events and offers many opportunities for future analyses concerning drought drivers and the propagation of droughts, including their possible self-propagation, as suggested by Schumacher et al. (2022). This approach could also help to link drought events



to their impacts more directly (Meadow et al., 2013; Lackstrom et al., 2017), as it spatiotemporally delimits drought-affected areas in great detail. Compared with similar preceding studies, this study is not based on the use of standard drought indices but rather on selected soil moisture variables from two different physically based models (SoilClim and mHM), which express the compound effects of temperature, precipitation, evapotranspiration and soil characteristics using land-surface and hydrological modeling, respectively. The presented method is robust enough for the delimitation of GLDEs and is very flexible in the selection of basic drought parameters, their thresholds and the use of various drought datasets; i.e., the method is applicable to different datasets and is repeatable.

The analysis results presented in Tables 1–4 show that the most important GLDEs during 1980–2022 occurred on all continents and appeared mainly after 2000. Although Sheffield et al. (2012) reported only a slight change in global drought (particularly based on the PDSI) during 1950–2008, and He et al. (2020) did not discover a worldwide increase in drought during the last two decades, many important drought events were reported worldwide after 2000 (e.g., Shmakin et al., 2013; Van Dijk et al., 2013; Griffin and Anchukaitis, 2014; Erfanian et al., 2016; Ionita et al., 2017; Marengo et al., 2017; Spinoni et al., 2017, 2019; Deng et al., 2020; Chiang et al., 2021; Moravec et al., 2021; Rakovec et al., 2022; Liu et al. 2023; Arias et al., 2024; Garrido-Perez et al., 2024) and were well reflected in the selected GLDEs in our paper (Tables 1–4 and S1–S2). Moreover, Fuentes et al. (2022) reported the intensification of drought characteristics in recent decades in several regions of the world (particularly in southern South America, central Australia, southwestern Africa, and central and eastern Asia). To demonstrate this situation, Table 5 shows a comparison of the absolute and relative numbers of GLDEs during 1980–2000 and 2001–2022. Except for category 1d in the dynamic classification, in all other categories for both classifications, the frequency of occurrence of the detected GLDEs was greater after 2000 than during the preceding period. According to the two-proportion Z test (Sprinthall, 2011), differences in the relative frequencies of given severity drought categories between two periods were statistically significant only for category 2s from SoilClim ($p < 0.05$), whereas for mHM, they were significant for 4s ($p < 0.05$), 2s and 5s ($p < 0.10$). For the dynamic drought categories, the related differences were statistically significant for category 5d from SoilClim ($p < 0.05$) and for categories 7d ($p < 0.10$), 4d and 6d ($p < 0.05$) from mHM.

**Table 5: Comparison of absolute (a) and relative (b; %) frequencies of global land drought events during 1980–2000 (A) and 2001–2022 (B) according to categories of severity (1s–7s) and dynamic (1d–7d) classifications from SoilClim and mHM. Statistical significance of relative frequencies: bold $p < 0.05$, italics $p < 0.10$.**

| Period | Severity classification | | | | | | | | | | | | | |
|---|---|---|---|---|---|---|---|---|---|---|---|---|---|---|
| | 1s | | 2s | | 3s | | 4s | | 5s | | 6s | | 7s | |
| | a | b | a | b | a | b | a | b | a | b | a | b | a | b |
| SoilClim model | | | | | | | | | | | | | | |
| A | 16 | 2.0 | **42** | **5.5** | 56 | 7.3 | 99 | 12.8 | 53 | 6.8 | 52 | 6.7 | 12 | 1.5 |
| B | 23 | 3.0 | **74** | **9.5** | 60 | 7.7 | 134 | 17.2 | 63 | 8.2 | 64 | 8.3 | 27 | 3.5 |
| mHM | | | | | | | | | | | | | | |
| A | 12 | 1.9 | *35* | *5.5* | 37 | 5.9 | **68** | **10.8** | *34* | *5.4* | 35 | 5.5 | 9 | 1.4 |
| B | 20 | 3.2 | *61* | *9.7* | 56 | 8.9 | **120** | **19.0** | *61* | *9.7* | 59 | 9.4 | 23 | 3.7 |




| Period | Dynamic classification | | | | | | | | | | | | | |
|--------|------|------|------|------|------|------|------|------|------|------|------|------|------|------|
| | 1d | | 2d | | 3d | | 4d | | 5d | | 6d | | 7d | |
| | a | b | a | b | a | b | a | b | a | b | a | b | a | b |
| SoilClim model | | | | | | | | | | | | | | |
| A | 17 | 2.0 | 46 | 5.5 | 54 | 7.2 | 107 | 12.8 | **37** | **6.8** | 57 | 6.7 | 12 | 1.5 |
| B | 22 | 3.0 | 70 | 9.5 | 63 | 7.8 | 125 | 17.3 | **79** | **8.2** | 59 | 8.3 | 27 | 3.5 |
| mHM | | | | | | | | | | | | | | |
| A | 16 | 2.5 | 35 | 5.6 | 39 | 6.2 | **68** | **10.8** | 36 | 5.7 | **28** | **4.4** | *8* | *1.3* |
| B | 16 | 2.5 | 59 | 9.4 | 57 | 9.0 | **119** | **18.9** | 59 | 9.4 | **66** | **10.5** | *24* | *3.8* |

Concerning the continental distribution of severe GLDEs, the lower relative proportions of categories 7s and 7d in Eurasia in both SoilClim and mHM are remarkable. This result could be influenced by the existence of one (SoilClim) or two (mHM) extremely large GLDEs during the last two decades in Eurasia (Tables 1–4), propagating particularly around Siberia

and the Central Asian Plains, where the potential for large dynamic events is the highest due to the absence of prominent topographic features. On the other hand, most of the other droughts that occurred during this period and were identified in other parts of Eurasia (southeast Asia or Europe), which has a greater amount of fractured topography, were smaller and shorter. Both the smallest absolute frequency and relative proportion of 7s droughts occurred in Australia. This finding is due to the fact that it is by far the smallest of the analyzed continents; however, in terms of the 7d category, the relative proportion

was much greater, which fits the geography of Australia, including a lack of larger mountain ranges and other elements, which could be expected to constrain drought propagation. With respect to the frequencies of 7s and 7d GLDEs, Eurasia and North America had the greatest absolute numbers of GLDEs, however South America had their highest proportion.

In the context of recent global climate change, increases in the frequency and intensity of drought events are among the most impactful changes worldwide. Our effort to catalog individual GLDEs during the last four decades at a high

spatiotemporal resolution could be effective and useful for benchmarking newly evolving droughts and contextualizing their potential impacts, while the developed methods can also be applied to new, emerging datasets with even greater accuracy. Moreover, as we consider droughts as fully 3D (area and time) events, our approach is designed to provide tools for analyses with the aim to connect drought occurrence with large-scale atmospheric circulation patterns and global climate variability modes; on the other hand, our approach can be employed to study processes leading to drought propagation (and possible self-

propagation) at regional to continental scales, as our spatial delimitations are not restrained by predefined regional borders.

## 6 Conclusion

From the severity and dynamic classifications of GLDEs applied to relatively available water in the soil from SoilClim model and to soil moisture from the mesoscale hydrologic model mHM from 1980–2022, our conclusions can be described as follows:

(i) The proposed GLDE classification method combines the spatiotemporal variability and dynamics of drought events with several characteristics that describe their extent, duration, severity and propagation. The method is robust for the delimitation of GLDEs and is very flexible for selecting basic drought characteristics and thresholds, as well as using basic drought datasets.

(ii) Using 10th percentile thresholds and clustering the gridded data in 10-day intervals for relative available water from SoilClim model and soil moisture from mHM, 775 GLDEs from SoilClim and 630 GLDEs from mHM were identified for 1980–2022. Based on the four severity characteristics and three dynamic characteristics, the identified drought events were divided into seven severity (from extremely weak to extremely severe) and seven dynamic (from extremely static to extremely dynamic) categories.

(iii) The distribution of the detected GLDEs to individual continents corresponds to their sizes. However, concerning the relative proportions, South America is prominent in terms of the extremely severe and extremely dynamic GLDEs, followed by North America for the extremely severe category, while Eurasia experienced the most extensive GLDEs according to both models.

(iv) The severity and dynamic categories of GLDEs show substantial overlap among the most severe categories, while the overlap mostly disappears in below-average categories. The most severe droughts seem to also be more dynamic; however, very small droughts can be not only static but also dynamic.

(v) The frequency of GLDEs generally increased during 2001–2022 compared to 1980–2000 across all drought categories. However, due to the high variability in the drought events, only some of our drought categories exhibited statistically significant increases.

**Data availability**

Global simulations of AWR/SM from SoilClim/mHM in 10-day interval and 0.5° resolution, which were used to create presented drought catalog are available from: https://doi.org/10.5281/zenodo.11395946 (Řehoř et al., 2024b). Lists of all GLDEs based on SoilClim and mHM data are in Word files in the Supplement.

**Author contributions**

JŘ: formal analysis, methodology, visualization, writing – original draft preparation. RB: conceptualization, methodology, writing – original draft preparation. OR: data curation, software, writing – review & editing. MH: conceptualization, methodology, writing – review & editing. MF: methodology, writing – review & editing. RK: conceptualization, methodology, writing – review & editing. JB: data curation, software. MP: writing – review & editing. VM: data curation. MT: conceptualization, methodology, writing – review & editing.



**Competing interests**

Luis Samaniego and Rohini Kumar are members of the editorial board of Hydrology and Earth System Sciences.

**Acknowledgments**

We acknowledge American Journal Experts for English style corrections.

**Financial support**

This research has been supported by the Ministry of Education, Youth and Sports of the Czech Republic (grant AdAgriF -
Advanced methods of greenhouse gases emission reduction and sequestration in agriculture and forest landscape for climate
change mitigation (CZ.02.01.01/00/22_008/0004635) and by Grant Agency of the Czech Republic for the project no. 23-
08056S "Dynamic tracking of drought events and their classification on the global scale – DynamicDrought".

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
