# Peer review of "Global catalog of soil moisture droughts over the past four decades"

_EGUsphere, 2024_

## Author Comment (AC2)

**RC2: Louise Mimeau**

The identification and classification of droughts events and the consideration of the spatial dynamics of events is both a novelty and an interesting contribution to the study of droughts at the global scale. I find it interesting that the authors have taken into account two different hydrological models to analyze the model uncertainty in their method. The paper is well written and can be suited for publication in HESS after a few major remarks are taken into account, mainly concerning a more in-depth discussion of the results.

Response: We would like to thank the reviewer for his in-depth assessment of our manuscript.

General Comments

I agree with the general comments of RC1 and I would like to add the following points in addition to his:

1. The paper would gain in clarity by regrouping parts 4.1 and 4.2 (the results obtained with SoilClim and mHM could be grouped together in the same figures and analyzed at the same time), and by discussing in greater depth the identifiaction of drought events with the OPTICS method (cf general comment n°2 from RC1). I also noticed in the supplementary materials that for each continent there are often distinct events with the same start and end dates (e.g. for the SoilClim model the events ranked 193 and 373 in S. America start at the end of April 1983 and end at the beginning of January 1984, or the events ranked 463, 532, 735 and 736 in N. America which all begin in August 2017 and end in November 2017). Can these events really be considered climatically distinct? What are the reasons for these distinctions between events? I think these questions should be addressed in the discussion section.

Response: The requested extension of discussion was done and is described below within the "line by line comments".

As we mentioned in reply to RC1, the aim of the article was to present enough robust methodology of GLDEs cataloging and document its employment for two independent data sources represented for both mHM and SoilClim models to demonstrate that the classification method works well for both models. The aim of the paper is not comparison of the two different model outputs. Therefore, we prefer to keep the models comparison separated by sections, except for Section 4.3 with Figure 15, which are specifically designated for this purpose. Moreover, merging SoilClim and mHM results into the same figures would make such resulting figures too extensive and complicated, same as their descriptions, that and particularly the describing text could confuse readers, regarding what concerns SoilClim and what mHM results.

Concerning the examples of GLDEs, occurring at the same time on the same continent, we acknowledge a certain level of uncertainty withing the clustering, which is inherent to any similar method. The algorithm clusters drought grids based on their spatiotemporal position and density-based clustering generally tends to merge areas with larger density into larger clusters and delimit more smaller clusters in areas with smaller density. It is not and cannot be the role of clustering to investigate whether drought in multiple nearby regions at the same time was caused by one large-scale climatic event or not. Looking for drivers of specific GLDEs could be a

subsequent analysis performed using the catalogue. However, the clustering ensures some level of spatial separation of those events. And whether the separation is too strict or too loose cannot be conclusively determined by any subjective or objective method. To make this clear to the readers, we disclosed this also in the supplementary material itself.

2. I have a few reservations about the severity and dynamics indicators, which in my opinion are too interdependent and may have an impact on the classification. I suggest to replace some of these indicators with new ones (see line by line comments below) or at least to discuss the implications of selecting these indicators further in the discussion.

Response: Based on droughts delimited by some criteria and OPTICS method of the grid-point clustering, we selected indicators, which in our opinion expressed our aim the best, i.e, to classify the drought events from the spatiotemporal and dynamic points of view by the most complex way (corresponding to model outputs). Each drought classification can be based on different criteria selected by corresponding authors (see e.g., Spinoni et al., 2019 or He et al., 2020), why other authors could select other indicators. To select really independent indicators for such events we see as extremely difficult or impossible task. Because this referee comment does not obtain proposal on such totally independent indicators, it is really difficult to add any others, because we selected the best ones according to our opinion.

Spatiotemporal characteristics of investigated GLDEs will always be to certain extent interdependent, based on basic facts concerning drought occurrence. Very short droughts generally cannot develop into continental-scale events or move to a completely different location. Therefore, a complete independence between used characteristics was not possible and not aimed for. However, we tried to include all main aspects of GLDEs within chosen characteristics. Reviewers specific comments concerning these characteristics are answered below within the "line by line comments" as well as the extension of the discussion section mentioned here.

3. Presenting some of the results in map form would help showing the severity and dynamics of the drought events. In particular, it would be interesting to show maps with the maximum spatial extent and the trajectory of the centroids for some of the most extreme events or for the events mentionned in the discussion section (e.g. L395, L402, L463).

Response: We understand the reviewer comment, but please consider that we are working in 10-day steps, which would mean that we should have 36 maps a year for one particular drought event. However, to illustrate the spatial nature of GLDEs, we created maps of the maximum extent and centroid movements of the three most extreme SoilClim-based GLDEs (the top 3 in Table 1) and an animation showing the development of the most severe GLDE during one year and included that all into the Supplement.

Line by line comments

L refers to line and P refers to page.

P3L74 : Please provide references for LAI, landuse and terrain inputs used for the modelling with SoilClim model.

Response: Accepted, the paragraph was extended as follows:

… SoilClim was applied to each grid with a daily input of meteorological variables that consisted of precipitation, temperature at 2 m above the ground, dew-point temperature at 2 m, wind speed at 10 m, and incoming shortwave radiation, which originates from ERA5-Land (Muñoz-Sabater et al., 2021), as well as with the leaf area index (LAI), land use and terrain inputs, also taken from the ERA5-Land dataset. The plant-available water capacity …

P3L86 : Why are the SoilClim and mHM models forced with two different datasets for the daily meteorological inputs (ERA5-Land and ERA5) ? Using the same inputs would make it easier to compare results obtained with SoilClim and mHM.

Response: The mHM is much more computationally demanding, therefore going for the higher resolution of ERA5-Land was not practical. Also it allowed slight perturbation in the modelling inputs as this particular study has not aimed at pure comparison of two different models i.e. SoilClim and mHM but to develop robust catalogue of soil drought episodes. It also needs to be noted, that the models were developed by different teams. However, both model outputs were rescaled to 0.5 ° resolution for the clustering process and as ERA5-Land is mostly just land-surface dynamic downscaling of ERA5, there is no reason to worry about huge differences in forcing which would manifest on this lower resolution.

P4L105 : Please provide the parameter values for the OPTICS clustering.

Response: In the the dbscan::optics() function in R, final parameters were: eps = 5, minPts = 20 and in dbscan::extractXi() function for cluster extraction, chosen parametr xi = 0.001.

P4L106 : How many clusters were removed from this filtering ? This information would be useful for analyzing the fraction of identified droughts that are local events versus the fraction of drought events on a broader continental scale.

Response: Accepted, following sentence was added into the manuscript:

… To eliminate regional cases with very small drought-affected areas, clusters that included fewer than 50 grids for one 10-day interval, fewer than 500 grids overall or that appeared in less than three 10-day intervals were excluded from both datasets, after which 775 clusters (further drought events) remained in the SoilClim dataset and 630 remained in the mHM dataset. In the case of both models, selected GLDEs comprise of around 15% of original clusters, however including over 80% of all grids inputted into the clustering. …

P5L148 : The relationships between the severity characteristics seem to be due to the fact that some the characteristics are inter-dependant. Characteristics b and d are directly related to c : the longer the event, the higher is the total sum of areal extents. b should perhaps be replaced with the averaged areal extent of a drought event during its duration and d with the average fraction of the drought events area with a AWR/SM value under the 2nd-percentile threshold.

Response: This problem we commented already in the response to your point 2. Here we add that severity characteristics were designed to represent spatiotemporal extent of GLDEs, i.e. their complete independence was not possible and not aimed for. Considering the reviewers

suggestion, it would basically mean dividing (b) and (d) characteristics by the (c) characteristic. However, having 2 out of 4 characteristics as mean values may downplay the impact that long-term drought events have on affected areas, which is why we used basically cumulative values for characteristics (b) and (d). And since this classification is specifically focused on spatiotemporal severity, we would prefer to keep the current characteristics.

Figure 4, 5, 6, 7, 11, 12, 13, 14, 15 : Please use different color scales for dynamic classification to avoid confusion with the severity classification.

Response: The 7-color scale was carefully prepared in accordance with the journal's policy to use CVD-friendly scales for all figures. We would prefer to keep it as it is, since the classifications are clearly distinguished by the category names (7s / 7d) and axes names, however if the reviewer or the editor insist, we will try to prepare a second 7-color scale that also sufficiently CVD-friendly.

P13L244 : Can the authors explain this relationship between categories S and D: why events with a wide spatial and temporal coverage are also the most dynamic? I believe this might due to the method used to calculate the dynamics characteristics, which are dependent on the duration of the event (especially for the indicator a: the longer the event, the greater the sum of the distances) and the spatial extent of the event (an event can be very extensive spatially and relatively static, but because of its spatial extent, a small shift in the centroid can give a large absolute distance). The dynamics characteristics should rather be computed as ratios between distances between the centroids and the width of the spatial exent (or number of grid cells) of the events and averaged per time interval.

Response: Characteristic (c) in the dynamic classification is "The mean geographic distance between centroid positions", so it is already averaged by time, as the reviewer suggests.

Characteristic (b) is also not cumulated over time, since it is (one) maximum distance between any couple of all centroid positions. So, if the centroid is e.g. just oscillating back and forth in a small area, this value remains low and will not cumulate over time. Of course, long-term droughts have a higher potential to actually move to a completely different location, but that is exactly one of the behavior patterns (maybe even the most interesting one) we wanted to comprehend by the dynamic classification.

So, this leaves only the characteristic (a) cumulating over time as the reviewer pointed out. However, even droughts that might experience some kind of "pulsing", or moving back and forth in a small area for a longer time period may be considered dynamic from a certain standpoint, compared to actually completely static droughts. Therefore, we would prefer to keep this characteristic in the dynamic classification.

Figure 7 and 14 : Please clarify axis labels (e.g. Severity category or Dynamic category)

Response: Accepted and corrected

Figure 15 : A clear separation between the 5 continents (a and b) and categories (c) would make the figure easier to read.

Response: Accepted and corrected.

P24L385 : Could changes in land cover or irrigation, which are not taken into account in the modelling, also be sources of uncertainty and have an impact on the classification of drought events?

Response: Yes, in some areas the landuse changes would be significant but rarely on the scale considered in the project. However, we added these factors into the list of uncertainties.

P24L390 : The authors should discuss in more detail the sensitivity of drought event identification to OPTICS clustering parameters.

Response: Accepted, following text was added into the paragraph:

… However, specific settings of the algorithm parameters must be partially derived empirically to fit the characteristics of a given dataset (particularly concerning its density) and prevent either the connection of all objects (in our case, grids in individual 10-day intervals) into one large cluster, including the whole dataset, or the failure of the algorithm to create clusters at all if the clustering parameters are too strict. Aside of these extreme cases, when the clustering failed, smaller changes of the parameters lead only to minor changes in the event delimitation. Still, there is no objective method for defining "perfect" parameters; hence, clustering uncertainty is inherent and affects the length of existence of individual clusters in our dataset. …

P26 Table 5 : To make the table easier to read, please replace A and B in the table respectively with 1980-2000 and 2000-2020, Response: Accepted and corrected

and perhaps just show the relative frequencies instead of showing both absolute and relative frequencies. Response: If in both reviews appeared several times request on comparison of both mHM and SoilClim models, then we see as important to preserve both absolute and relative frequencies in this table in order to demonstrate differences or agreement between these two datasets.

P26L455 : It should be pointed out in the discussion that this statistical analysis over two 20-year periods is a little short-sighted for identifying trends (especially when some drought events can last several years).

Response: Accepted, following text was added into the paragraph:

To demonstrate this situation, Table 5 shows a comparison of the absolute and relative numbers of GLDEs during 1980–2000 and 2001–2022. As these periods are from a climatological point of view relatively short, the comparation should be taken with caution. Except for category 1d in the dynamic classification, …

 L553, L590, L651, L663, L699 : Some doi or url are missing in the references.

Response: Accepted and corrected.

L964 : A line break is missing before Vincente-Serrano et al, 2022

Response: Accepted and corrected.

---

## Author Response (AR1)

**Referee Comment 1:**

Summary

This paper analyses global soil moisture drought using data derived from the SoilClim and mHM models from 1980 to 2022. A drought catalog was compiled based on severity and dynamic drought classifications utilizing a threshold approach and the OPTICS clustering technique. The authors identified hotspot regions for extremely severe and extremely dynamic drought events in the South American region and North America. The longest and most extensive droughts occurred in Eurasia. The authors highlighted an increase in global land drought events over the past decade. Furthermore, they also suggest that this study may serve as a basis for future investigations into drought drivers and impacts.

Assessment

This paper presents a new technique to analyze drought classification based on severity and dynamic classifications. The manuscript is interesting and well written. I have a few minor comments below and three general comments, but only for clarification and improvement. I believe this work is well suited for publication in HESS.

Response: We would like to thank the reviewer for his in-depth assessment of our manuscript.

General Comments

I have two general comments regarding the manuscript but all of them are only for clarification, suggestion, and improvement of the manuscript.

I am curious as to why did the authors combine Europe and Asia continents into a single region called Eurasia? This combination results in a significantly high number of Global Land Drought Events (GLDEs) in this region, as depicted in Figures 3, 6, and 10. Furthermore, as a reader, I would prefer to see separate results for Europe and Asia. I suggest splitting the Eurasia results into distinct sections for Europe and Asia. This approach would likely be more engaging for readers from these two continents.

Response: We understand that our use of Eurasia as one continental area is an unusual or not so frequently used approach. We understand, that division of Europe and Asia might be rooted very deep from political, economic, societal and cultural standpoints, but from the standpoint of physical geography the border is practically non-existent and just divides artificially a large continental area in two parts. Moreover, during the data processing we found out, that there is a surprisingly large number of GLDEs with large areas in both Europe and Asia, particularly among the largest GLDEs. The area around the geographical border of Europe and Asia, particularly Eastern Europe, Central Asian and Western Siberia, comprises of extensive plains, almost without any orographic barriers and therefore it is ideal for both formation and movement of large GLDEs. From these reasons, the division of these GLDEs between Europe and Asia for the statistical summaries seems to be rather artificial, so we would like to keep more correctly the entire area as one "continent".

I am confused about the ongoing drought occurrences from November 2004 to December 2022 in Table 1, which spans over 18 years. How did you analyze this multi-year, prolonged drought duration in Eurasia? I believe this period likely consists of multiple drought events in Europe and Asia occurring from November 2004 to December 2022 in different regions. Please correct me if I am mistaken. Additionally, how does this analysis account for the European drought of 2003?

Response: The SoilClim model results delimited this GLDE as continuously existing area of drought, moving particularly through Siberia, Eastern Europe, Central and Eastern Asia without ceasing to exist for the entire period since November 2004. It needs to be noted that it was not always extremely large or intensive, but it fulfilled our basic conditions for its delimitation and it had multiple "peaks" during the period 2004–2022. The mHM results are not entirely different in this instance, however, it defined two (still quite long) GLDEs, namely in June 2013 to December 2022 and from April 2007 to July 2014 (Table 1). Concerning a deeper analysis you mention – this is actually a next step we are planning to take in the future, when we want to focus on these few extremely large, intensive and dynamic events and investigate them further. However, this is something that could not be fitted into this manuscript, which serves a purpose of presenting our methods of GLDEs delimitation and their basic statistical overview. The European drought of 2003 appears in our catalogue but considering the global point of view and selected criteria, it did not place at a top position within our classification.

In this paper, the authors present many figures but the explanation about the findings is limited. For example, Figures 1 and 2 are described in a single paragraph, whereas a more detailed discussion of the findings would be beneficial. In Figure 1, why is the scatter in (a)-(c) greater than in (b)-(c)? Similarly, in Figure 4, why is the relationship in (a)-(b) more linear than in the other panels? Additionally, I believe it would be more useful to provide the average findings from both the mHM and SoilClim models.

Response: Accepted, to follow your recommendations for more detail discussion of figures, we added further description as follows:

… Some plots of pairs from the four characteristics in Fig. 1 display the grouping of GLDEs into lines because there are many GLDEs with the same value (meaning the same score assigned to them) in the case of a single-digit number of 10-day intervals. The largest scatter is present in the duration and maximum extent relationship (Fig. 1(a)–(c)), which shows that drought with longer duration does not necessarily have to reach very large extent. On the other hand, the relationship between duration and total extent is much stronger, as it is a cumulative value. The described features of the scatterplots in Fig. 1 are confirmed in the boxplots of these categories,…

… Compared to the four characteristics of severity classification (Fig. 1), they show more consistent patterns with more concentrated fields of related points, particularly for categories that include GLDEs with average movements (category 4d) to extremely static droughts (1d). The very strong relationship between maximum and total centroid distance proves that GLDEs with long trajectory are usually not just oscillating around the same area but actually move through continents. The box plots of these categories are shown in Fig. 5, …

Concerning the "average findings" from mHM and SoilClim results, the aim of the article was to present enough robust methodology of GLDEs cataloging and document its employment for two independent data sources represented for both mHM and SoilClim models to demonstrate that the classification method works well for both models. However, based on comments from both reviewers and the editor, we decided to merge Sections 4.1 and 4.2 into one section describing both mHM and SoilClim results together. Moreover, we also merged Figures 1–14 into just Figures 1–7, united as couples showing always the same variable for both SoilClim and mHM. The findings are therefore now much more "condensed" and easier comparison between the two models is possible.

Line by line comments

L refers to line and P refers to page.

P2L41: Maybe write examples of traditional meteorological drought indices, e.g., SPI and SPEI?

Response: Accepted, the paragraph was extended as follows:

… To understand the spatiotemporal variability and severity of droughts, meteorological drought indices (e.g., Standardized Precipitation Index – SPI, Standardised Precipitation-Evapotranspiration Index – SPEI or Palmer Drought Severity Index – PDSI) have traditionally been used (e.g., Spinoni et al., 2014, 2015, 2019; Chiang et al., 2021; Fuentes et al., 2022; Vicente-Serrano et al., 2022), …

P3L87: The evapotranspiration method should be mentioned.

Response: Accepted, the paragraph was extended as follows:

… The daily minimum, maximum, and mean temperatures are also used to obtain potential evapotranspiration estimates (Hargreaves and Samani, 1985), for which the Penman-Monteith method as described by Allen et al. (1998) was used. Our simulations are based …

P4L100-101: Please write what is D2 and what is S2 for readers who are not working with the US drought monitor and Czech drought monitor, respectively.

Response: Accepted, the names of both categories used within the drought monitors were added as follows:

… Drought occurrence has been identified using the 10th-percentile drought, which is in line with using this threshold in US Drought Monitoring (Svoboda et al., 2002) since 1995 for the "D2" category ("Severe Drought") definition and in the Czech Drought Monitor System (Trnka et al., 2020; Intersucho, 2024) since 2012 as the "S2" ("Moderate Drought") category. To assess the most severe drought, the 2nd-percentile drought (i.e., 50-year return period) was calculated using the same approach. …

P4L115: What do the authors mean with the total sum of areal extent of 2nd percentile drought? Why not 10th percentile as well?

Response: The 10th percentile serves as our basic threshold for drought selection. However, we wanted to include information about intensity of given GLDE into the classifying process. Therefore, we calculated the extent of area of even much more extreme drought, based on 2nd percentile (return period of 50 years), within the every 10th-percentile-based GLDE and used it as fourth characteristic in classification.

P5L150: Maybe replace the word "are variable" with "vary"?

Response: Accepted and corrected.

P6: Figure 1. Maybe make the legend (colored circles) bigger? Also I prefer to label all figures with letter a, b, c, and so on.

Response: Accepted, we made the legend larger.

P7: Figure 2. What are upper and lower quartile? Are they 75th and 25th percentiles?

Response: Accepted, we changed the caption of Figure 2, to … median, 75th and 25th percentiles, maximum and minimum…

P8: Please use comma to write number million in all tables. For example: 6,701,638.

Response: Accepted and corrected.

P12L227-228: Here the authors mention about centroid movements. However, I could not see these values in the Table 2. These values should be written in the Table or somewhere.

Response: The centroid movements are represented in Table 2 by the "Dynamic scores", corresponding to the movement characteristics defined in Section 3.3

P17L295: The authors may move the word "(Fig. 11)" to the end of sentence.

Response: Accepted and corrected.

P24L380: Maybe provide references about the uncertainties of SoilClim model.

Response: Accepted, following publications were added as reference for SoilClim uncertainties:

… In the SoilClim model, there are uncertainties related to the schematization of individual processes, soil profile vertical discretization, and other scale-related assumptions (e.g., each grid cell is represented by dominant landcover), described in detail by Hlavinka et al. (2011), Trnka et al. (2020) and Řehoř et al. (2021). …

P24L397: The authors could explain the potential alternatives for drought clustering as proposed by the cited literature.

Response: Accepted, the paragraph was extended as follows:

... Potential alternatives for drought clustering have been proposed for example by Andreadis and Lettenmaier (2006), Vidal et al. (2010) or Samaniego et al. (2013), including stepwise selection

of contiguous drought areas, k-means cluster analysis or Density-based spatial clustering of applications with noise (DBSCAN), which is alternative to OPTICS, using similar approach. …

P25L402-403: Here the authors mention drought event in North America, which appeared around the center of the continent. However, I could not see any Figure showing the centroid of the drought events. Why don't the authors provide this figure in the appendix or supplementary material?

Response: We added a figure depicting all centroid positions of this event into the supplementary material and mentioned in the discussion as follows:

… Such an example could be the drought event from June 2014–March 2015 in North America (Table 4), which appeared around the center of the continent between other, more consolidated clusters. All centroid positions of this GLDE are shown in Figure S5. A few problematic cases such as this represent some uncertainty in the classification integrity. …

P26: Table 5. Maybe in this table the authors can provide the average results of SoilClim and mHM.

Response: Because both model outputs lead to a different number of GLDEs, simple averaging of these data in terms of long-term trends seems not to be suitable or representative from methodological point of view.

**Referee Comment 2:**

The identification and classification of droughts events and the consideration of the spatial dynamics of events is both a novelty and an interesting contribution to the study of droughts at the global scale. I find it interesting that the authors have taken into account two different hydrological models to analyze the model uncertainty in their method. The paper is well written and can be suited for publication in HESS after a few major remarks are taken into account, mainly concerning a more in-depth discussion of the results.

Response: We would like to thank the reviewer for his in-depth assessment of our manuscript.

General Comments

I agree with the general comments of RC1 and I would like to add the following points in addition to his:

1. The paper would gain in clarity by regrouping parts 4.1 and 4.2 (the results obtained with SoilClim and mHM could be grouped together in the same figures and analyzed at the same time), and by discussing in greater depth the identifiaction of drought events with the OPTICS method (cf general comment n°2 from RC1). I also noticed in the supplementary materials that for each continent there are often distinct events with the same start and end dates (e.g. for the SoilClim model the events ranked 193 and 373 in S. America start at the end of April 1983 and end at the beginning of January 1984, or the events ranked 463, 532, 735 and 736 in N. America which all begin in August 2017 and end in November 2017). Can these events really be

considered climatically distinct? What are the reasons for these distinctions between events? I think these questions should be addressed in the discussion section.

Response: Accepted, original Sections 4.1 and 4.2 were merged into one section describing both mHM and SoilClim results together. Moreover, we also merged Figures 1–14 into just Figures 1–7, united as couples showing the same variable for both SoilClim and mHM. The findings are therefore now much more "condensed" and easier comparison between the two models is possible.

Concerning the examples of GLDEs, occurring at the same time on the same continent, we acknowledge a certain level of uncertainty within the clustering, which is inherent to any similar method. The algorithm clusters drought grids based on their spatiotemporal position and density-based clustering generally tends to merge areas with larger density into larger clusters and delimit smaller clusters in areas with smaller density. It is not and cannot be the role of clustering to investigate whether drought in multiple nearby regions at the same time was caused by one large-scale climatic circulation pattern or phenomena or not. Looking for drivers of specific GLDEs could be a subsequent analysis performed using the catalogue. However, the clustering ensures some level of spatial separation of those events. And whether the separation is too strict or too loose cannot be conclusively determined by any subjective or objective method. To make this clear to the readers, we disclosed this also in the supplementary material itself.

Moreover, the requested extension of discussion was done and it is described below within the "line by line comments".

2. I have a few reservations about the severity and dynamics indicators, which in my opinion are too interdependent and may have an impact on the classification. I suggest to replace some of these indicators with new ones (see line by line comments below) or at least to discuss the implications of selecting these indicators further in the discussion.

Response: Based on droughts delimited by some criteria and OPTICS method of the grid-point clustering, we selected indicators, which in our opinion expressed our aim the best, i.e. to classify the drought events from the spatiotemporal and dynamic points of view by the most complex way (corresponding to model outputs) and broad aspects and definitions of droughts. Each drought classification can be based on different criteria selected by corresponding authors (see e.g., Spinoni et al., 2019 or He et al., 2020), why other authors could select other indicators. To select really independent indicators for such events we see as extremely difficult or impossible task. Because this referee comment does not obtain proposal on such totally independent indicators, it is really difficult to add any others, because we selected the best ones according to our opinion.

Spatiotemporal characteristics of investigated GLDEs will always be (to a certain extent) interdependent, based on basic facts concerning drought occurrence. Very short droughts generally cannot develop into continental-scale events or move to a completely different location. Therefore, a complete independence between used characteristics was not possible and not aimed for. However, we tried to include all main aspects of GLDEs within chosen characteristics. Reviewers specific comments concerning these characteristics are answered

below within the "line by line comments" as well as the extension of the discussion section mentioned there.

3. Presenting some of the results in map form would help showing the severity and dynamics of the drought events. In particular, it would be interesting to show maps with the maximum spatial extent and the trajectory of the centroids for some of the most extreme events or for the events mentionned in the discussion section (e.g. L395, L402, L463).

Response: We understand the reviewer comment, but please consider that we are working in 10-day steps, which would mean that we should have 36 maps a year for one particular drought event. However, to illustrate the spatial nature of GLDEs, we created maps of the maximum extent of the three most extreme SoilClim-based GLDEs (the top 3 in Table 1) and an animation showing the development of the most severe GLDE during 2019 and included that all into the Supplement (S3 and S4). It is also mentioned in the manuscript as follows:

… maximum extent of 5.9 million km2 in 2022. The maximum extent of the three most extreme GLDEs based on SoilClim model is shown in Figure S3. Concerning the mHM-based analysis …

… a defined multidimensional space (in our case, three-dimensional). To illustrate the spatiotemporal behavior of the clusters, we included animation of their development during 2019 to Supplement S4. However, specific settings …

Line by line comments

L refers to line and P refers to page.

P3L74 : Please provide references for LAI, landuse and terrain inputs used for the modelling with SoilClim model.

Response: Accepted, the paragraph was extended as follows:

… SoilClim was applied to each grid with a daily input of meteorological variables that consisted of precipitation, temperature at 2 m above the ground, dew-point temperature at 2 m, wind speed at 10 m, and incoming shortwave radiation, which originates from ERA5-Land (Muñoz-Sabater et al., 2021), as well as with the leaf area index (LAI), land use and terrain inputs, also taken from the ERA5-Land dataset. The plant-available water capacity …

P3L86 : Why are the SoilClim and mHM models forced with two different datasets for the daily meteorological inputs (ERA5-Land and ERA5) ? Using the same inputs would make it easier to compare results obtained with SoilClim and mHM.

Response: The mHM is much more computationally demanding, therefore going for the higher resolution of ERA5-Land was not practical, while it was possible perform do SoilClim simulations in higher resolution. However, as Muñoz-Sabater et al. (2021) and Cucchi et al. (2020) clearly state, ERA5-Land is just a land-surface dynamic downscaling of ERA5, with mostly linear interpolation of ERA5 fields on near-surface meteorological variables into a higher

resolution grid, so there is no reason to worry about differences in forcing, particularly because both model outputs were rescaled to 0.5 ° resolution for the clustering process, so even half the resolution of original ERA5. Moreover, both models have been developed by different teams over long periods of time and more detailed descriptions of the models, including all parameters and chosen inputs reasoning can be found within multiple cited publications from Sections 2.1 and 2.2.

P4L105 : Please provide the parameter values for the OPTICS clustering.

Response: In the the dbscan::optics() function in R, the final parameters were: eps = 5, minPts = 20 and in dbscan::extractXi() function for cluster extraction, chosen parameter xi = 0.001.

P4L106 : How many clusters were removed from this filtering ? This information would be useful for analyzing the fraction of identified droughts that are local events versus the fraction of drought events on a broader continental scale.

Response: Accepted, following sentence was added into the manuscript:

… To eliminate regional cases with very small drought-affected areas, clusters that included fewer than 50 grids for one 10-day interval, fewer than 500 grids overall or that appeared in less than three 10-day intervals were excluded from both datasets, after which 775 clusters (further drought events) remained in the SoilClim dataset and 630 remained in the mHM dataset. In the case of both models, selected GLDEs comprise of around 15% of original clusters, however including over 80% of all grids inputted into the clustering. …

P5L148 : The relationships between the severity characteristics seem to be due to the fact that some the characteristics are inter-dependant. Characteristics b and d are directly related to c : the longer the event, the higher is the total sum of areal extents. b should perhaps be replaced with the averaged areal extent of a drought event during its duration and d with the average fraction of the drought events area with a AWR/SM value under the 2nd-percentile threshold.

Response: We commented on this issue in the response to reviewer's point 2. Here we can add that severity characteristics were designed to represent spatiotemporal extent of GLDEs, i.e. their complete independence was not possible and not aimed for. Considering the reviewer's suggestion, it would basically mean dividing (b) and (d) characteristics by the (c) characteristic. However, having 2 out of 4 characteristics as mean values may downplay the impact that long-term drought events have on affected areas, which is why we used basically cumulative values for characteristics (b) and (d). And since this classification is specifically focused on spatiotemporal severity, we would prefer to keep the current characteristics.

Figure 4, 5, 6, 7, 11, 12, 13, 14, 15 : Please use different color scales for dynamic classification to avoid confusion with the severity classification.

Response: The 7-color scale was carefully prepared in accordance with the journal's policy to use CVD-friendly scales for all figures. We would prefer to keep it as it is, since the classifications are clearly distinguished by the category names (7s / 7d) and axes names, however

if the reviewer or the editor insist, we will try to prepare a second 7-color scale that also sufficiently CVD-friendly.

P13L244 : Can the authors explain this relationship between categories S and D: why events with a wide spatial and temporal coverage are also the most dynamic? I believe this might due to the method used to calculate the dynamics characteristics, which are dependent on the duration of the event (especially for the indicator a: the longer the event, the greater the sum of the distances) and the spatial extent of the event (an event can be very extensive spatially and relatively static, but because of its spatial extent, a small shift in the centroid can give a large absolute distance). The dynamics characteristics should rather be computed as ratios between distances between the centroids and the width of the spatial exent (or number of grid cells) of the events and averaged per time interval.

Response: Characteristic (c) in the dynamic classification is "The mean geographic distance between centroid positions", so it is already averaged by time, as the reviewer suggests.

Characteristic (b) is also not cumulated over time, since it is (one) maximum distance between any couple of all centroid positions. So, if the centroid is e.g. just oscillating back and forth in a small area, this value remains low and will not cumulate over time. Of course, long-term droughts have a higher potential to actually move to a completely different location, but that is exactly one of the behavior patterns (maybe even the most interesting one) we wanted to comprehend by the dynamic classification.

So, this leaves only the characteristic (a) cumulating over time as the reviewer pointed out. However, even droughts that might experience some kind of "pulsing", or moving back and forth in a small area for a longer time period may be considered dynamic from a certain standpoint, compared to actually completely static droughts. Therefore, we would prefer to keep this characteristic in the dynamic classification.

Figure 7 and 14 : Please clarify axis labels (e.g. Severity category or Dynamic category)

Response: Accepted and corrected

Figure 15 : A clear separation between the 5 continents (a and b) and categories (c) would make the figure easier to read.

Response: Accepted and corrected.

P24L385 : Could changes in land cover or irrigation, which are not taken into account in the modelling, also be sources of uncertainty and have an impact on the classification of drought events?

Response: Yes, in some areas the landuse changes could be significant but rarely on the scale we considered in this study. However, we added these factors into the list of uncertainties as follows:

… We also acknowledge that the fixed/prescribed model functions representing hydrological processes might not be equally plausible across different parts of the world. Long-term changes

in land cover and irrigation can also cause larger uncertainties on a regional scale. Some uncertainties are associated with the clustering technique …

P24L390 : The authors should discuss in more detail the sensitivity of drought event identification to OPTICS clustering parameters.

Response: Accepted, following text was added into the paragraph:

… However, specific settings of the algorithm parameters must be partially derived empirically to fit the characteristics of a given dataset (particularly concerning its density) and prevent either the connection of all objects (in our case, grids in individual 10-day intervals) into one large cluster, including the whole dataset, or the failure of the algorithm to create clusters at all if the clustering parameters are too strict. Aside of these extreme cases, when the clustering failed, smaller changes of the parameters lead only to minor changes in the event delimitation. Still, there is no objective method for defining "perfect" parameters; hence, clustering uncertainty is inherent and affects the length of existence of individual clusters in our dataset. …

P26 Table 5 : To make the table easier to read, please replace A and B in the table respectively with 1980-2000 and 2000-2020, Response: Accepted and corrected

and perhaps just show the relative frequencies instead of showing both absolute and relative frequencies. Response: As in both reviews a request for comparison of both mHM and SoilClim models appeared several times, we see as important to preserve both absolute and relative frequencies in this table in order to demonstrate differences or agreement between these two datasets.

P26L455 : It should be pointed out in the discussion that this statistical analysis over two 20-year periods is a little short-sighted for identifying trends (especially when some drought events can last several years).

Response: Accepted, following text was added into the paragraph:

To demonstrate this situation, Table 5 shows a comparison of the absolute and relative numbers of GLDEs during 1980–2000 and 2001–2022. As these periods are from a climatological point of view relatively short, the comparison should be taken with caution. Except for category 1d in the dynamic classification, …

L553, L590, L651, L663, L699 : Some doi or url are missing in the references.

Response: Accepted and corrected, doi/url added if it was available.

L964 : A line break is missing before Vincente-Serrano et al, 2022

Response: Accepted and corrected.

**Editor decision:**

Thank you very much for your proposal on how to address the reviewers' comments and suggestions. I would like to invite you to revise the manuscript as outlined in your response. In addition, I would like to encourage you to pay more attention to the following points, which I consider to be important and think have not yet received sufficient attention in your responses:

Response: We would like to thank the editor for considering our manuscript. We tried to do our best and we revised our original responses to the referees and further revised the manuscript.

(1) Drought definition: consider potential adaptations to avoid identification of overly long drought events (R1)

Response: We extended our explanations to R1 and R2 concerning the clustering method, discussed possible clustering alternatives within the discussion and added Supplements S3–S5 to better illustrate spatiotemporal features of our clustering method.

(2) Avoid redundancy and reduce number of figures: Think about how to combine results from two models to avoid redundancy and shorten the manuscript (4.1 and 4.2, R1 and R2)

Response: Accepted, original Sections 4.1 and 4.2 were merged into one section describing both mHM and SoilClim results together. Moreover, we also merged Figures 1–14 into just Figures 1–7, united as couples describing the same variable for both SoilClim and mHM. The findings are therefore now much more "condensed" and easier comparison between the models is possible.

(3) Choice of indicators for clustering (R2): consider the reviewer's suggestion more seriously

Response: We are very sorry about the editor's opinion, that we did not take the reviewer's suggestion (R2) more seriously. We know very well, how droughts, their characteristics and delimitation represent very complex and complicated problem. As follows from different global drought classifications (e.g., He et al., 2020; Monjo et al., 2020), selection options for basic parameters are very broad, and the parameters are not always all independent. We tried to present one robust method of spatiotemporal and dynamic drought cataloging. If other authors use this method in future, they may use different parameters considered by them as important. That is also our case: we selected parameters we considered the most important. That allowed us to present one specific drought catalog, which is of course dependent on the parameter selection, definition of Global drought events (GLDEs), clustering method, etc. But the methods we present in the manuscript allow future authors to try some different parameters and then, as expected, to obtain slightly different catalog.

(4) Avoid inconsistent use of climate forcing (R2)

Response: We further clarified that as stated by Muñoz-Sabater et al. (2021) and Cucchi et al. (2020), ERA5-Land is just a land-surface dynamic downscaling of ERA5, with mostly linear interpolation of ERA5 fields on near-surface meteorological variables into a higher resolution grid, so there is no reason to worry about differences in forcing, particularly because both model outputs were rescaled to 0.5 ° resolution for the clustering process, so even half the resolution of original ERA5. Moreover, both models have been developed by different teams over long

periods of time and more detailed descriptions of the models, including all parameters and chosen inputs reasoning can be found within multiple cited publications from Sections 2.1 and 2.2.

---

## Author Response (AR2)

**Referee report 1:**

Assessment

I appreciate the authors' careful consideration of my main concerns in their revised manuscript and improved the clarity of the paper. I accept the manuscript with only very minor revision, mainly textual.

Response: We would like to thank the reviewer for his valuable and thorough assessment of our manuscript.

Line by line comments

L refers to line and P refers to page.

P2L38-39: In this sentence, suddenly the authors mention compound droughts and heatwaves. I am thinking maybe just write heatwaves as an example? "In this context, the compound effects of droughts with e.g., heatwaves have……"

Response: Accepted and corrected.

P3L89-90: I would like the authors to check carefully again about the ET method. The reference seems for Hargreaves method while the authors stated PM method. I think you should remove reference Hargreaves.

Response: Reviewer is correct, and we apologize for the confusion. In this work, the mHM model uses the Hargreaves-Samani method; therefore, we removed the second part of the sentence: ""

P6L159: Fig. 1(a)-(c) maybe write it Fig. 1A(a)-(c) like you did for the rest of the figure references.

Response: Accepted and corrected.

P8L178: You cite Figure 3Aa twice.

Response: Accepted and corrected.

P9L198: It is not eight because N. America drought occurred in July 1999 so it was before 2000.

Response: Accepted and corrected.

P9L200-201: Explanation in the rebuttal should be written here so the readers understand that the long droughts were not occurred only in one location.

Response: Accepted, we added an explanation as follows:

"The most extreme identified drought, which began in November 2004 in eastern Eurasia, was still ongoing until December 2022 and achieved a maximum extent of 6.7 million km2 in 2021 and was moving through particularly Eastern Europe and Sibiria during this period, changing its location and extent."

P10: Table 1: could you check all the alphabets for the scores in all tables? Some use small letter some use capital letter. For example, Table 2 for C, table 3 for B and A, etc.

Response: Accepted and corrected.

P21L365: Maybe explain again what is c.

Response: Accepted, "c." changed to "approximately".

**Referee report 2:**

The article brings a specific approach to global evaluation of the drought in a certain statistical categories at global level and it differs from the theoretical framework of generalized drought of smaller areal dimension. The positive value of the calculated drought characteristics in this work is the concept of "consecutive" occurrences of water deficiency. This approach brings a valuable view in theory on drought categorization at global scale and as stated in the abstract it gives a good opportunity to analyze the evolving features of spatiotemporal linked drought events. On the other hand it does not have concrete practical application and it would be difficult to investigate the impacts of such defined drought events as the impacts are of rather smaller dimension with strong spatial variations depending on the local conditions. An innovative approach is presented in using two model systems as well as a specific approach in analyzing the results. However, I have several questions that should be answered in the paper and several formal comments that should be addressed.

Response: We would like to thank the reviewer for his/her valuable and thorough assessment of our manuscript.

We would just like to note that while this study, (conducted on a global scale) could not study drought impacts on a regional scale, we believe our catalog has an application potential as the delimited and classified GLDEs on individual continents can be investigated in terms of impacts in following studies.

General notices

The severity of the drought is defined in a special way depending on the area affected by the drought. Drought as understood in this study results from a deficiency of water in surface or subsurface components of the hydrologic system expressed in AWR and SM terms. Its start is defined by a statistical percentile limit while the severity is attached to the areal extent of the drought within its duration. This approach expresses a certain level of the water deficiency in the area but it does not effectively take into account the aggravating effect behind the defined statistical threshold. This calculation allows to express a certain intensity threshold and its duration but not the actual intensity of the drought and its variation within this time interval I terms of water deficiency level. Intensity drought balance is an important component of drought mainly with the respect to long lasting (even years) drought which is limited by applying above described approach. The areal extent does not express the real deficit of water. This fact should be discussed in text.

Response: The selected definition of drought allows for objective and repeatable determination of drought event start, duration and also intensity since within the analyzed parameters both maximum extent and total time x area affected has been considered in clustering the events. It should be stressed that the approach is primarily used to catalogue drought events and categorize them to allow for testing hypothesis with regards to impact of individual events using e.g. status

of vegetation data, wildfire extent or reported drought impacts on stream flow, reservoir water levels or yields. Moreover, we would like to mention that our severity classification uses the intensity parameter defined by lower percentile to account for occurrence of very severe water deficit withing the defined "drought area".

To clarify that our approach does not study the impacts of sustained drought in one location, we extended the discussion as follows:

"… We also acknowledge that the fixed/prescribed model functions representing hydrological processes might not be equally plausible across different parts of the world and our approach also does not aim to study impacts of sustained drought in one location. …"

Introduction of the dynamic factor in terms of drought geographical spread and motion gives very valuable dimension to drought evaluation. Nevertheless, the dynamic characteristic expressed by the centroid positions and their movement is in a certain way interconnected with the spatiotemporal characteristics of drought severity and these two characteristics are not fully independent one from another. All the centroids, thought their occurrence is time dependent, must necessarily be located within the area defined in the severity categories.

Response: Looking for connections between drought severity (particularly concerning its extent) and dynamic characteristics is one of the reasons why we created the presented catalog. It is logical, that the extent of a drought event is interconnected with its movement and evolution in time generally, and our approach enables basically quantify and compare these drought event aspects. So, we do not see this interconnection as a negative thing.

The soil water supply in presented work is basically based on the precipitation only. Nevertheless, the continental dimension of the drought evaluation includes the vast regions supported by water from surrounding mountains both by surface and underground flows. The earth surface in this study is taken rather as a flat area from this point of view (what is in a contradiction to the statement regarding the drought propagation in Australia). This should be mentioned or discussed in the text, as well.

Response: We appreciate the reviewer's comment on the potential role of lateral subsurface flows into soil water supply in hydrological processes, especially in mountainous regions. We fully recognize the importance of subsurface flow contributions at local to regional scales, we would like to emphasize that our study considers soil moisture at a spatial resolution of $0.5°$ (50 km at the equator). Sub-grid topographic features and local-scale heterogeneities (such as hillslopes and small catchments that typically drive lateral subsurface flow) are not explicitly resolved at this coarser resolution. Therefore, the dominant hydrological processes captured at this scale are those governed by large-scale atmospheric inputs, like precipitation, as the reviewer correctly pointed out, as well as the surface processes (e.g., evapotranspiration and runoff), with soil moisture dynamics primarily influenced by vertical fluxes. Many previous large-scale hydrological and land surface modelling studies (Telteu et al 2021 or Müller Schmied et al 2024) have similarly assumed that lateral subsurface flow can be neglected at resolutions of $0.25°–1°$, as its integrated effect tends to be minimal when averaged over larger grid cells.

Telteu, C.-E., et al. (2021): Understanding each other's models: an introduction and a standard representation of 16 global water models to support intercomparison, improvement, and communication, Geosci. Model Dev., 14, 3843–3878, https://doi.org/10.5194/gmd-14-3843-2021

Müller Schmied, H., et al. (2024): Graphical representation of global water models, EGUsphere [preprint], https://doi.org/10.5194/egusphere-2024-1303

Specific notices

Data – why are the databases for SoilClim calculation and mHM calculation different even in the precipitation and temperature parameters ERA5-Land (Muñoz-Sabater et al., 2021) vs ERA5 reanalysis (Hersbach et al., 2020)?

Response: We acknowledge the reviewer's point regarding the different meteorological inputs (ERA5-Land vs. ERA5) used in the SoilClim and mHM model setups. These setups were developed independently by separate teams prior to this study. While we recognise the potential for differences arising from the use of distinct datasets, it's important to note that our primary objective is not a direct comparison of SoilClim and mHM, but rather the creation of a robust soil drought catalogue. As ERA5-Land is essentially a land-surface enhanced downscaling of ERA5, the differences in temperature and precipitation fields are relatively small, particularly when aggregated and averaged out to the 0.5 ° spatial resolution at the end, as we used in our analysis. This aggregation process effectively minimizes the influence of fine-scale variations. Therefore, we think that the minor discrepancies between ERA5-Land and ERA5 do not significantly affect the large-scale drought patterns identified in our catalogue.

-S1 and S2 reflect the differences in methodology; but how far the differences in databases can also contribute to these differences?

Response: As mentioned in the previous response, ERA5/ERA5-Land forcings differ only marginally at the coarser (spatially aggregated) resolution. However, there is no reason why this difference should make significant changes on a continental scale. The differences between GLDEs in S1 and S2 are primarily the result of the difference between SoilClim and mHM.

-R86 "This study considers soil moisture (SM) simulations averaged over the entire 2 m soil column depth (aggregating values over six soil layers) to quantify shallow water availability". How far is SM comparable with AWR?

Response: AWR is basically relative soil moisture but considers only values between the wilting point (0%) and the field capacity (100%). In the case of soil moisture below/above these thresholds, AWR just stays 0/100%. SM is "standard" relative soil moisture defined as the ratio between volumetric soil moisture and maximum available soil water capacity, so it has a wider scale than AWR, but otherwise both variables do not differ. Nevertheless, we would like to highlight that for the purposes of our analysis, both AWR and SM were subsequently transformed into percentile-based indicators. This transformation effectively normalized the two variables, which allowed for a direct comparison of their relative dryness or wetness across different spatial locations and periods, regardless of their original scale. Therefore, while AWR

and SM differ in their raw representation of soil moisture, their percentile-based transformations allow for meaningful comparisons within our study's context.

-The severity scores in Table 1 should be better defined/clarified. Drought occurrence was identified using the 10th-percentile drought. In Table 1 (a) represents maximum areal extent of calculated out of 10th percentile; (d) represents drought intensity expressed as the total sum of areal extents of 2nd-percentile drought; but (d) is equal and even bigger than a) in table 1. This is confusing and should be better explained/clarified in text or in Table 1 legend.

Response: The scores are explained in detail in Section 3.2:

"… Subsequently, for each of these characteristics and both datasets, the identified drought events (775 for SoilClim and 630 for mHM) were placed in order from the lowest to the highest values of the given characteristic, and orders of the values were used as scores. For example, a score of 1 was attributed to the event with the lowest value of the given characteristic, and a score of 775 or 630 was attributed to the event with the highest value of this characteristic for SoilClim and mHM, respectively."

So, the scores represent just an order of the given GLDE in given characteristic, which was calculated separately for each characteristic. Therefore, the fact that values of characteristic (a) are inherently higher than characteristic (d) does not reflect into the scores, which range 1–775 (or 1–630 for mHM) for all four characteristics.

The explanation of scores is too long to be put into Table 1 caption, however we clarified the caption as follows:

"… according to the severity classification. The severity scores calculation is described in Sect. 3.2 from severity characteristics which are specified in points a–d in Sect. 3.2 (* indicates ongoing droughts)."

- P11, R 227-28 The claim „The box plots of these categories are shown in Fig. 5, ... with nearly no overlap in ... categories doesn`t look to be the full truth – This claim is not fully visible in (A) soil clim (b) and (c) as well as in (B)mMH (b) and (c) in Fig. 5.

Response: Accepted, we adjusted the sentence as follows:

"The box plots of these categories are shown in Fig. 5, which reveals that the employed characteristics decrease in a stepwise manner from the category of extremely dynamic droughts (7d) to that of extremely static droughts (1d), with only small overlaps in values among the interquartile ranges of the seven individual categories."

-P20 -there is a number of uncertainties originating from the different databases, methodological performances and clustering described. Nevertheless, there is no estimate of any of these uncertainties. Percentages The level of elimination of the effect of uncertainties that affect the long-term means within individual grids should be documented or at least discussed in some example(s).

Response: Using both SoilClim and mHM independently in the entire study is our main approach to demonstrate the level of uncertainty and how the results change with a different approach. However, we cannot exactly quantify the magnitude of uncertainties for most variables such as meteorological inputs.

-P20, R 344-346 Table S2 does not show the disconnection of GLDE event but the overlap in 2013/06/26 – 2014/07/10.

Response: There is a temporal overlap but these two GLDEs are not spatiotemporally connected. The second GLDE started while the first one was diminishing in a different location, so the algorithm defined them as separate but in the case of SoilClim based GLDEs there was a spatial connection and a very long-lasting GLDE was delimited.

P23, R419-421 "…Australia, including a lack of larger mountain ranges and other elements, which could be expected to constrain drought propagation". This is questionable statement which should be supported by some arguments regarding Australian climatography. The influence of orography o the drought spread is region specific and the influence of atmospheric circulation must not be neglected in such evaluation.

Response: Accepted, we acknowledge that our statement was inaccurate, and we changed the sentence (deleting our speculation regarding the geography of Australia) as follows:

"This finding is due to the fact that it is by far the smallest of the analyzed continents; however, in terms of the 7d category, the relative proportion was much greater."

- As no impacts of any drought event listed in Tab 1 and 2 was mentioned the whole study has to be taken as a purely theoretical approach to evaluate the spread and dynamic of drought in a certain relative categories. There are some unspoken simplifications of the influence of geographical features of the earth's surface and the effect of atmospheric circulation was not considered. Nevertheless, the article provides a comprehensive robust methodology for global drought assessment and can serve as a basis for further investigation of identified the drought events and their propagation at regional and global scale.

Response: This study primarily presents our approach to delimit, catalog and classify drought events. We believe this catalog will be used further for the study of drought impacts and drought drivers such as atmospheric circulation, however we did not see any more space within this study to include these aspects (which are thematically out of the main aim of the study) as the study is rather extensive as it stands.

Formal errors

-Capital D in Table 1 indicating the drought intensity (should be small d). Similarly in Table 2 C->c.

Response: Accepted and corrected.

-No keywords list available

Response: We used the suggested "Copernicus_Word_template" that does not include keywords.

-P3, R90 Allen et al. (1998) used in text is not listed in References

Response: Accepted and corrected.

-P21, R353 Table 4 is not available neither n the manuscript or i the supplements

Response: Accepted and corrected to Table 2B.

-P21, R359 Fig. 11 is not available neither n the manuscript or i the supplements

Response: Accepted and corrected to "Figs. 1 and, 4".

-Table 3 – there are both capital and small letters used to mark the absolute and relative frequencies in severity classification section of the table

Response: Accepted and corrected.